# Dynamics of specialization in neural modules under resource constraints

Gabriel Béna ⬤ ✉ & Dan F. M. Goodman ⬤ ✉

The brain is structurally and functionally modular, although recent evidence has raised questions about the extent of both types of modularity. Using a simple, toy artificial neural network setup that allows for precise control, we find that structural modularity does not in general guarantee functional specialization (across multiple measures of specialization). Further, in this setup (1) specialization only emerges when features of the environment are meaningfully separable, (2) specialization preferentially emerges when the network is strongly resource-constrained, and (3) these findings are qualitatively similar across several different variations of network architectures. Finally, we show that functional specialization varies dynamically across time, and these dynamics depend on both the timing and bandwidth of information flow in the network. We conclude that a static notion of specialization is likely too simple a framework for understanding intelligence in situations of real-world complexity, from biology to brain-inspired neuromorphic systems.

Modularity is an enticing concept that naturally fits the way we attempt to understand and engineer complex systems. We tend to break down difficult concepts into smaller, more manageable parts. Modularity has a clear effect in terms of robustness and interpretability in such systems. Disentangled functionality means that the impairment of a module doesn't lead to the impairment of the whole, while making it easy to spot critical failure points. When conjoined with redundancy, modularity thus forms the basis of an extremely robust general organizational principle[1]. Following the early observations of Brodmann[2] on cytoarchitecture, it has been thought to be an essential element in the organization of the brain: brain organs can be found structurally and should have distinct functional roles. Brodmann himself was no naive localizationist when it came to the brain, admitting that only very elementary functions could be completely localized, and that complex brain functions would only emerge through the interaction of many of those sub-organs[3]. Nevertheless, a fairly modular view of the brain has since prevailed. However, the precise link between structure and function has never been conclusively established, and recent work has begun to challenge this organizational principle[4,5].

As a more emergent principle linking structure and function, it has been suggested that modularity emerged as a byproduct of the evolutionary pressure to reduce the metabolic costs of building, maintaining and operating neurons and synapses. It has been shown that

brain networks are near-optimal when it comes to minimizing both its wiring and running metabolic costs. For example, the summed length of the wiring in the brain has been minimized[6], with nodes being near-optimally placed[7–10]. At the same time, brains also display complex and efficient topology allowing for incredible information processing capabilities. To do so brain networks possess some very costly elements to maintain such as long-range connections, a 'rich-club' of densely connected neurons, and some non-optimal placement of components[11–14]. Based on these observations, brain organization could be understood as a trade-off between mitigating these inherent metabolic costs (from being a spatially embedded network) and its efficiency and capabilities of information processing (needed to solve complex tasks and display the emergent intelligence that we witness)[15].

We thus have to distinguish two types of modularity, *structural* and *functional*, and understand how they are related. We take structural modularity to mean the degree to which a neural network is organized into discrete and differentiated modules. Such modules show much denser connections internally than across other modules. This is usually computed using the Q-metric[16], measuring how much more clustered these modules are when compared to a graph connected at random. Although note that many other techniques are possible, and that module detection in networks[17] as well as defining measures of modularity[18,19] are complex and interesting fields in their

Imperial College London, London, UK. ✉e-mail: g.bena21@imperial.ac.uk; d.goodman@imperial.ac.uk

own right. While this structural definition is important, it doesn't necessarily inform us on the function of the modules.

Functional modularity refers to the degree to which potential modules—which may or may not be the same as structural modules—perform specialized and distinct functions. Potential definitions have been put forward, to try to conceptualize what should constitute modules. While the details have been contested, to frame our definitions it may be helpful to recall some of the properties of Fodor's definition[20] refined in[21]. This can help us pinpoint a few properties that a functionally modular system should possess, and allow us to formulate associated quantitative measures.

1. Modules should only carry a sub-function of a more complex overall function.
2. Domain specificity states that a module should respond to and operate only on specific types of inputs.
3. Separate modifiability means that the impairment of one module should not affect the functioning of another.
4. Information encapsulation asserts that a module should have restricted access to information outside its own state.

An important property, hinted at by these definitions, that we do not address directly in this study is the ability of a network to recompose previously acquired (and encapsulated) knowledge to achieve systematic generalization. We touch on this important feature in the discussion.

In animal brains, functional modules are usually identified in two ways: by using data from neural activity or by studying the impact of lesions. The former requires large-scale (or even whole-brain) recordings, and consequently relies on examining co-activations of brain areas, based on a proxy measure. This would relate to point (2)—*domain specificity*. In the human brain, fMRI measures changes associated with blood-flow to infer neural-activity, that then serves as a basis to infer functional connectivity[22–24]. In small vertebrates like zebrafish, that can take the form of calcium imaging[25–27]. Nevertheless correlations can only tell us so much (as Ralph Adolphs and David Anderson would put it, imaging a car's speedometer to infer its speed wouldn't tell us much about the underlying mechanisms moving it forward[28]).

Possibly the most authoritative way of examining the functional organization of the brain relies on lesion analysis. Studies examine naturally occurring injuries (in humans at least) to understand the effect that the impairment of an area has on behavior. This would relate to the concept of *separate modifiability*. Those lesions can however be hard to come by—even if it has been argued that a systematic collaboration across hospitals could provide a substantial lesion database[29]. Moreover lesion analysis needs consistent cross-validation and control to correctly infer a functional link. It has been shown that single-lesion experiments could be insufficient to properly infer causation, or even be biased[30–32], and that this includes double dissociation studies relying on single-lesions injuries or manipulations[33]. Nevertheless, ablation-based analysis remains a central tool for understanding neural networks, both biological and artificial[34]. Overall, gaining a rigorous causal understanding of brain functions seems to be a highly challenging problem. While all the aspects discussed above can help us conceptualize and measure specialization, they would also be challenging to clearly test in real intelligent systems. We've seen that an analysis of both activity and lesions could prove insufficient to provide a full functional understanding. This is where experiments on artificial neural networks (ANNs) could prove crucial, as both can be precisely and simultaneously applied (on the same networks), and where other aspects like points (1) and (4) can be investigated.

A better understanding of the concepts of modularity in silico could result in fresh insights when it comes to natural brains for two reasons. First, ANNs have increasingly been proven to be valuable models of brain functions. Learning in both artificial and biological neural networks has to solve many of the same problems (like credit assignment: the problem of understanding how an individual element or parameter contributes to the performance of a network). Although the precise mechanisms used are likely to be different it has been suggested that the solutions could end up relatively similar, as biological brains could be approximating exact gradients in a local manner[35,36]. Spiking neural networks (SNNs) have been developed, relying on a plausible way of communication between neurons, and implementing biological learning mechanisms, neuronal models, and general organization[37]. ANNs have also proven remarkably able to predict[38] and decode[39] patterns of neural activity in the brain. Finally, some strong patterns of similarity have been observed in the activity of biological and artificial neural networks (such as receptive fields or grid-cells neurons)[40,41]. This solidifies the idea that studying modularity in artificial networks could refine our understanding of it in the brain (as in, for example[42,43]).

Second, ANNs are fully and precisely controllable, and are in that sense the perfect testing ground to study structural and functional definitions of modularity. They can act as a sort of baseline, allowing us to discard measures and definitions that fail to give coherent results even in a perfectly controlled environment. Not only that, but ANNs increasingly incorporate modularity explicitly in their design, making the study of this concept especially relevant even in such an abstract setting. Recent deep learning models are modular in the sense that they are compositions of a few types of basic building blocks. However, it is not clear that they are functionally modular in the way the brain is thought to be[44]. Moreover, ANNs usually disregard the fact that biological networks are spatially embedded networks running under constraints, a feature we discussed earlier as a potential key to understanding neural network organization. Nevertheless, work has been put into developing a new general framework explicitly incorporating modularity into models, and the field is actively gaining momentum and attention[45,46].

To allow us to tease apart the factors influencing the relationship between structure and function, we designed flexible artificial neural networks performing tasks in a highly controlled synthetic environment. Generally, the link between structural and functional modularity is context-dependent and involves a complex and dynamic interplay of several internal and external variables. However, it is unclear the extent to which structural modularity is important for the emergence of specialization through training. We show here a case where even under strict structural modularity conditions, modules exhibit entangled functional behaviors. We then explore the space of architectures, resource constraints, and environmental structure in a more systematic way, and find sets of necessary constraints (within our constrained setup) for the emergence of specialized functions in the modules. Finally, by measuring specialization in networks unrolled in time, we find that specialization actually has complex temporal dynamics governed by the dynamics of the inputs and communication between modules. Refining our conceptual understanding of modularity and specialization in this abstract setting may lead to fresh insights into the role of modularity in the brain and in brain-inspired algorithms and neuromorphic hardware.

## Results
### Experimental setup
To systematically investigate the relationship between structural and functional modularity, we develop a flexible artificial neural network (ANN) framework with precise control over network architecture and task design. Our approach focuses on quantifying the emergence of functional specialization under varying structural constraints, addressing a critical gap in understanding neural network organization. We design a flexible recurrent neural network (RNN) architecture and challenge it with a carefully constructed parity classification task

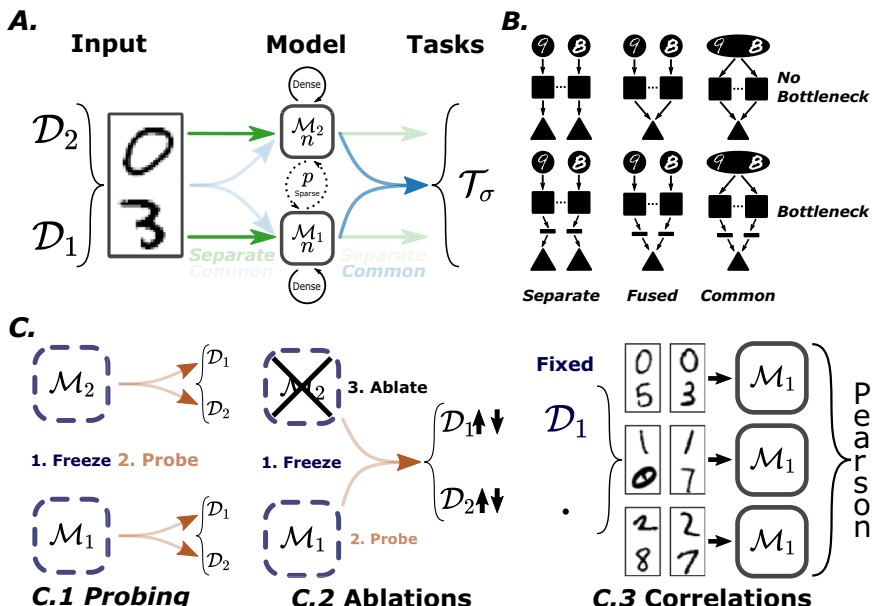

**Fig. 1 | Summary of methods. A** Schematic of input, model and task. Two digits $\mathcal{D}_1$ and $\mathcal{D}_2$ are fed to the network via either shared input weights (in which case each module sees each digit) or separate (in which case each module sees only a single digit). Two recurrent neural network modules of size $n$ then process the inputs, and communicate via a sparse interconnection containing a fraction $p$ of all $n^2$ possible connections. Outputs are computed either through shared readout (both modules feed into the same readout) or separate (in which case each module has its own readout, and the decision is based on a max function). A bottleneck consisting of an additional layer of 5 neurons can be placed between the modules and the final readout. **B** All possible architectural choices. **C** The three functional specialization metrics: Module Probing, where the trained network is frozen and a separate (probing) network is trained to extract digit identity from each module's hidden state; Module Ablation, where the trained network is frozen and a single network is trained to extract digit identity from both modules' hidden states, after which one is ablated and impact on performance is measured; and Correlation Analysis, where hidden states are correlated holding one digit fixed and varying the other.

using MNIST digits and EMNIST letters (see "Environment: data and tasks"). The network architecture allows for variable module sizes ($n$), sparse inter-module connectivity ($p$), and alternative input-output pathway configurations (see "Networks"). Having control over this $p$ connectivity also directly controls the structural modularity of the model, quantified by the widely used Q metric (see "Structural modularity"). While structural modularity has been extensively studied, functional modularity is less well understood, necessitating the development of robust metrics to quantify functional specialization. To this end, we develop three complementary metrics (see "Metrics"): Module Probing, which evaluates a module's ability to classify digit identity; Module Ablation, which measures performance loss upon module masking; and Hidden State Correlations, which analyses module representations under fixed digit input. This multi-faceted approach enables us to investigate the complex interplay between structural organization and functional specialization in artificial neural networks. Methods are summarized in Fig. 1 and more explicitly detailed in "Methods".

## Moderate structural modularity is not a sufficient condition for specialization

We wish to investigate whether imposing structural modularity is enough to guarantee specialization. To test this idea in a simple and controllable setting, we use an architecture composed of two densely recurrent modules, each receiving its own input (image of an MNIST digit) and producing its own outputs (the *separate-pathway* architecture from Fig. 1B). The global task requires each module to recognize the digit presented to it, but also depends on the parity of the other digit. It can therefore be broken down into two sub-tasks (digit recognition) that are able to be solved by the modules individually, but where (idealizing somewhat) solving the global task requires the modules to share just one bit of information from the solution of their sub-task (as it depends on whether or not the parity of the two digits is

the same or different). Modules consist of 25 neurons, a regime where the network can solve the task, without being over-parameterized (a parameter choice we will understand in more detail in the next section). Then by varying the fraction of active connections $p$ in the communication layer between modules, we can directly control the structural modularity of the model. For dense intra-module connectivity and a proportion $p$ of active inter-module connections, the standard measure of structural modularity can be calculated as $Q = (1 - p)/2(1 + p)$ (see "Structural modularity"). Test accuracy on this task increases with $p$ (unsurprisingly given that this means more parameters), but is always above chance and below saturation (Supplementary Fig. 1).

We measured the level of specialization of networks with varying inter-connection sparsity $p$ using three metrics (see "Metrics") that capture different aspects of functional specialization (see "Introduction"). We find that even at levels deemed very modular ($Q > 0.4$), the model only shows weak specialization (Fig. 2A). We can see how sharply the metrics rise when approaching maximal $Q$ values, suggesting extreme levels of structural modularity are needed for specialization to emerge. The relationship is clearer as a function of $p$ on a logarithmic axis, with specialization naturally dropping when approaching dense communication (Fig. 2B). This lets us draw some initial conclusions: in at least one simple type of network, imposing moderately high levels of structural modularity doesn't directly lead to the emergence of specialized modules. Moreover, the metrics we defined show similar trends, indicating that we are indeed capturing something meaningful. For the remainder of this paper, we present results mainly using the module probing metric.

## Environmental structure and resource constraints determine specialization

We have seen that a high level of structural modularity isn't a sufficient condition for the emergence of specialization. We now systematically

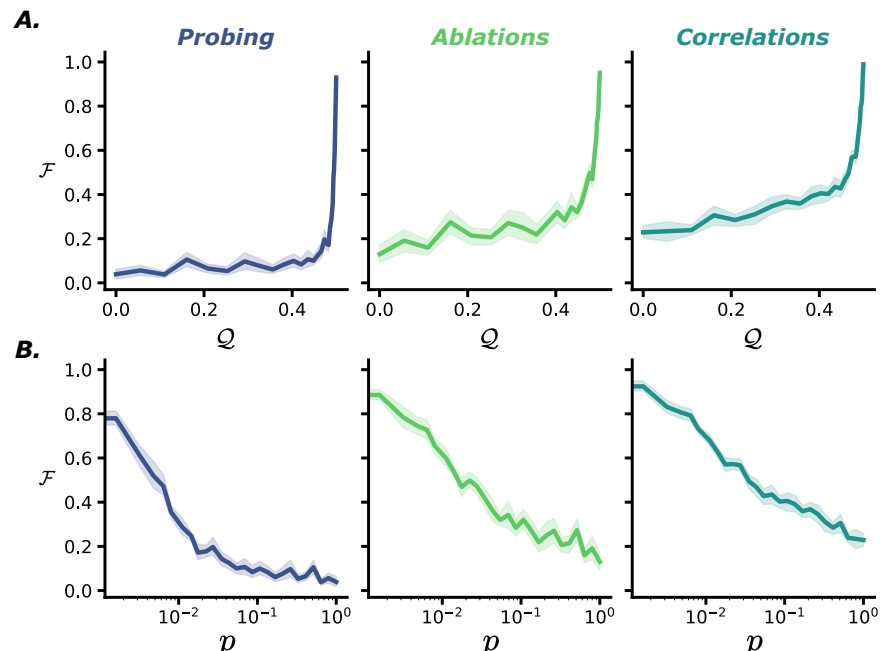

**Fig. 2 | Relationship between functional and structural modularity.** We plot levels of functional specialization of networks with (**A**) varying modularity $Q$, or equivalently (**B**) Inter-module connectivity $p$, measured by three different metrics (columns). All metrics indicate a similar trend, sharply rising only at extreme levels of $Q$ modularity. Data are presented as mean values with a shaded standard error envelope.

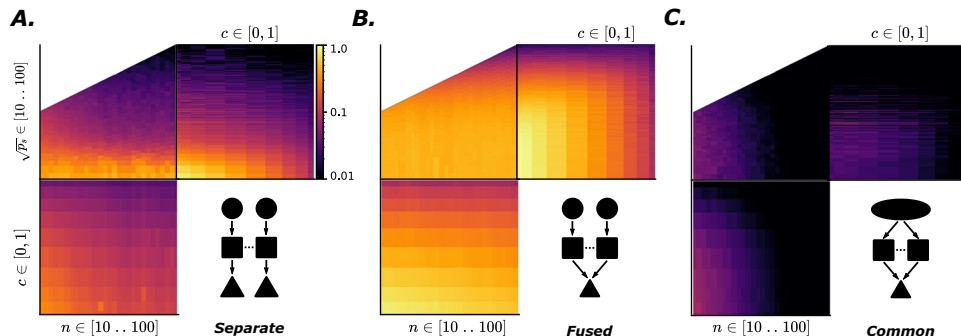

**Fig. 3 | Functional specialization of networks with varying module size *n*, number of active synapses in the communication layer $p_s$, in an environment where input variables present covariance *c*.** Each image shows functional specialization when varying two of these parameters (averaged across the third one), with color indicating the degree of functional specialization on a log scale, cut off at a mimal value of 0.01. Results are shown for three different architectures: (**A**) separate pathways, (**B**) fused pathways, and (**C**) shared pathway.

analyze the effect on specialization of the choices made in the architecture and environment of our toy model, to understand why and how specialization does emerge. We find that three main categories of effects interact to influence the emergence of specialization.

First are environmental effects: In the previous section, we explicitly designed the data to have a perfectly separable structure. However, features are rarely perfectly independent in real environments (a horse is more likely to have a green background, while a car is more likely to have a gray background). To model these complex dependencies in a simple way, we vary the covariance matrix of the distributions of digits between sub-tasks, essentially modifying the probability of seeing the two digits being equal, $\mathcal{D}_1 = \mathcal{D}_2$. We find that high covariance $c$ prevents specialization (Fig. 3 bottom left, or top right of each sub-figure), as knowing one hidden variable is already highly informative of the other. In Supplementary Fig. 3, we can also see that when the inputs to the two modules are drawn from separate datasets (as is the case when training with E-MNIST "Environment: data

and tasks") modules show stronger specialization. This would be the case in a multimodal setting, such as audio-visual integration. It makes sense that when sub-tasks have different input distributions, this should promote specialization, since the knowledge of decoding one wouldn't transfer as much to the other. Moreover, the computational capacity needed to decode both would be higher when sub-tasks are different in nature, changing some of the metabolic cost effects we discuss later on.

Second we look at metabolic cost constraints: To understand how resource constraints could shape the structure-function relationship of neural networks, we vary the number of elements that are costly to maintain or manufacture (in terms of both energy and space): neurons in a module ($n$) and synapses between modules (proportion $p$ of $n^2$ possible, or total number $p_s = pn^2$). We then measure the resulting specialization of the networks, when pushed towards their minimal values, lead to specialization arising, but the extent of which depends strongly on the architectural choices (as discussed in

the next point). We can also see that the precise interactions between environmental ($c$) and metabolic ($n$, $p$) variables strongly depend on the choice of the model architecture (Fig. 3).

Third, we investigate architecture choices: The effect of architecture choice is more subtle. A few salient points include:

1. Input scheme: We find that having separate input pathways constitutes a strong structural prior, leading to higher functional modularity, whereas having a shared input pathway lowers it substantially (Fig. 3A/B vs Fig. 3C). This intuitively makes sense, as modules are forced to "work with what they've got". When presented with different inputs, the only source of collapse of specialization comes from the information leakage introduced by the interconnections. As such, the main factor driving specialization, in that case, is the number of synapses $p_s$ (Fig. 3A top left). Nonetheless, having a shared input does not prevent the appearance of a specialization pattern, albeit at a reduced scale. We find that in that case, the significant parameter is the size of modules, ie the number of neurons they're composed of (Fig. 3C top left). Modules are forced to specialize when arriving at the threshold at which a single module does not have the computational capabilities to predict both digits (even when having both as input).

2. Bottleneck: Imposing a narrow bottleneck before the modules' output also lead to higher specialization overall (Supplementary Fig. 3). This may be because introducing a bottleneck on the output of the modules limits the possible computational role of the output layer itself, and forces the modules to do the majority of the work, which is a prerequisite for seeing specialization in the modules.

3. Output scheme: With separate outputs, the modules compete for which module determines the global readout through a max function. In that case we see a quantitatively lower specialization (Fig. 3A vs Fig. 3B). This could be explained by the *decision-making* dynamics of modules competing for the readout (a simplified demonstration of this effect is showed in Supplementary Fig. 2). In this structural configuration, networks often resort to having a main *decision-making* module. Although the task is designed to be solvable in a modular fashion, having such a module take all decisions means it's able to predict both digits and thus displays less specialization, when it comes to our metrics at least. One could nevertheless argue that resorting to such a separation of function (decision-making vs context-providing modules) is in its own right a form of specialization, albeit one that is not picked up by our metrics. Finally, having a common readout seems to have a coordinating effect leading to modules being more specialized across the board, alleviating the necessity of extreme resource constraints to see the emergence of specialization.

Results from the complete parameter sweep are shown in Supplementary Fig. 3.

### Function specialization can vary dynamically

Functional specialization is usually implicitly understood as a static property, i.e. a module is functionally specialized or not. Here, we investigate this assumption by quantifying specialization at every time step (Fig. 4), in the *fused pathway* architecture. We find that when presented with static inputs, the time course of specialization mainly depends on the timing of inter-module communications (Fig. 4A). When communicating at high bandwidth ($p$), the specialization of modules drops significantly and immediately after modules start to communicate. This collapse happens immediately, with no continual leakage over time, i.e. no further reduction of specialization after the initial drop. We hypothesize that this is a result of the static nature of inputs, where all the information is readily available at every time step and so there is less advantage to ongoing communication after the

initial burst (the only benefit of maintaining active communication is to free up working memory resources).

To confirm this intuition based on the static nature of inputs, we introduce dynamic structured noise (pseudo-code provided in Supplementary 1.5.1). With a noisy input, additional information about the underlying signal is introduced at each time step, giving the network an extra reason to engage in ongoing communication. We use two possible noise levels (Fig. 4B). This time we observe a continual decrease of specialization over time, in highly interconnected modules (i.e. the metric keeps dropping after the initial drop). This is especially true when decoding highly noisy versions of the inputs, meaning there is an increased advantage to ongoing communication.

Finally, we introduce stochasticity in the input dynamics by having individual digits be turned on from a purely noisy input at random times (pseudo-code provided in Supplementary 1.5.2), but with inter-module communication always on (Fig. 4C). We find that specialization dynamics follows input dynamics. When neither digit is switched on, neither module specializes (first time-step of every sub-plot). Once digit 0 (arrow upward) is turned on, module 0 starts to specialize on that digit, and similarly for digit 1 (downward arrow) and module 1. When only one digit is turned on, the other module also starts to specialize on that digit in the absence of any other input to specialize on. Once both digits are switched on, modules start by specializing in their respective digits, before the eventual drop in specialization over time. Specialization is more fluid in more densely connected networks, moving towards the non-primary digit faster in the absence of a primary digit input, and also decaying to zero faster once both digits are present (as in previous results). Intuitively, this can be summarized as specialization following the total amount of information received from the two digit sources (since sparser connectivity means lower bandwidth).

## Discussion

We used a carefully controlled toy network to investigate the relationship between structural and functional modularity. In doing so, our first challenge was to define modularity. For structural modularity, we simply used the popular graph theoretic $Q$ metric[16]. For functional modularity, or specialization, there is no similar consensus. Our first main result is that we found that three different metrics inspired by classical definitions of modularity gave broadly compatible results in our setup, suggesting that even though functional modularity is difficult to define and the subject of some contention, there may be a meaningful underlying concept. However, we also found that there is a dynamical aspect which is not captured in these definitions of functional modularity and which may turn out to be critical to understanding information flows in more complex networks and environments. Functional specialization should be understood in terms of states, or trajectories, with modules varying in their specialization depending on input and module communication dynamics. This supports recent developments in neuroscience that have started to rethink the causal narrative behind brain functions in a more dynamical manner and how it is related to a modular organization[47,48], for example looking at temporal contributions of elements in a system[49].

Our second main result is that imposing some degree of structural modularity on a network is not, in general, sufficient to induce functional specialization, even in a toy setting seemingly ideal for the emergence of specialization. We conclude that (a) in machine learning and neuromorphic computing applications, if we wish to encourage the emergence of functional specialization, structural modularity could prove insufficient, and (b) in neuroscience, if we observe structural modularity, we should not necessarily conclude that these modules will be functionally specialized. One limitation on this conclusion is that we have relied on the well-established $Q$-metric from network theory. This metric is widely used in connectomics research,

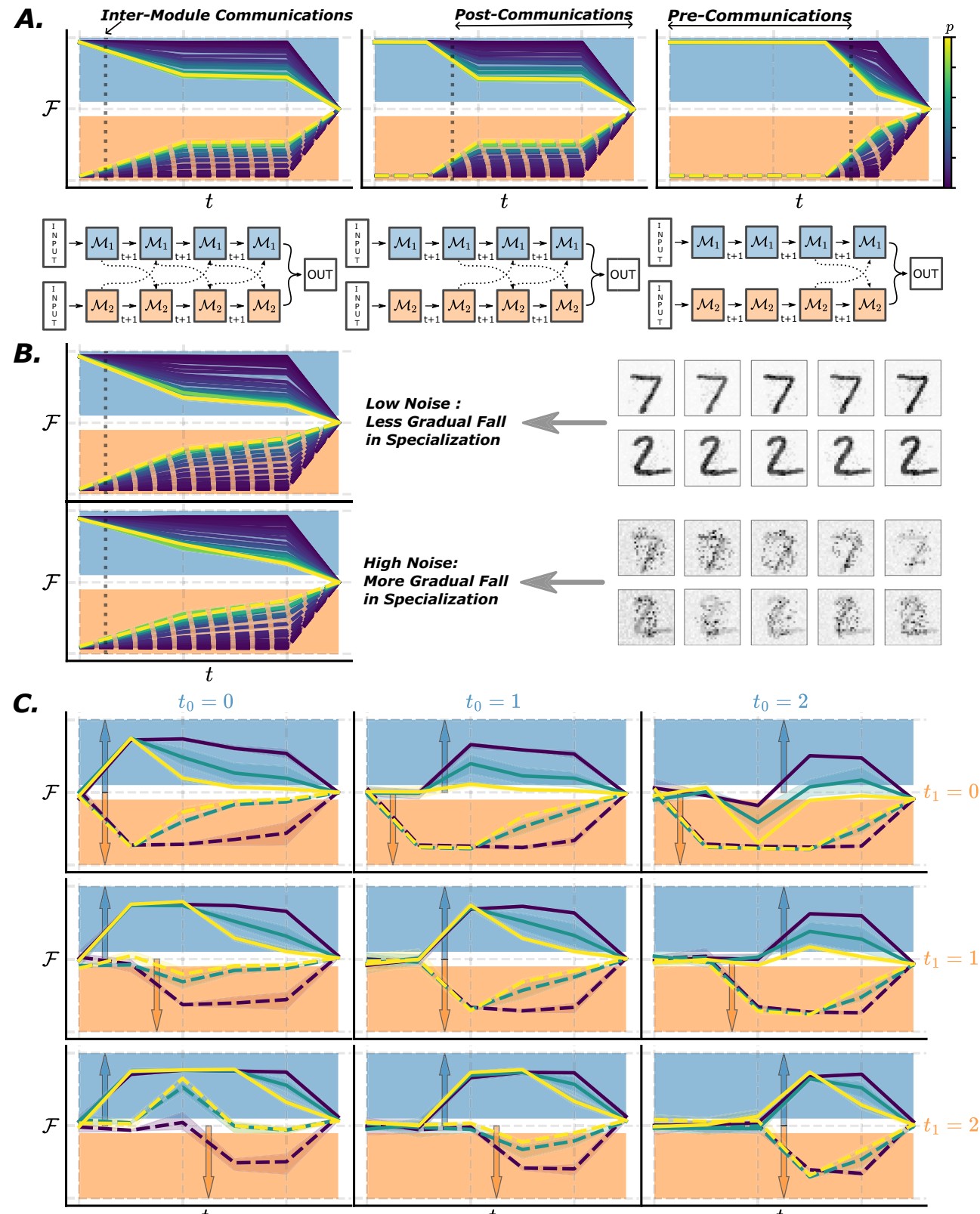

**Fig. 4 | Specialization dynamics. A** Specialization of different parts of the network at different time-steps. The network unrolled through time is shown below, with inputs presented to modules at time $t = 0$ at the left hand side, and subsequent time steps moving towards the right finishing in the readout layer at the final time step. In the plot above is shown the specialization of the corresponding module at a given time, where specialization close to 1 means full specialization on digit 0, close to -1 means full specialization on digit 1, and close to 0 means no specialization. The dark dashed lines in both the network diagram and the plot show the timing of inter-module communication. Each colored line represents networks with different inter-connection levels $p$. We vary the communication timings in the inputs, and find specialization collapses when communicating at high bandwidth ($p$ high). **B** With dynamic noise in the inputs, specialization drops continually, especially in high noise setting. **C** With stochasticity in the input dynamics (digits 0 and 1 turned on at $t_0$ and $t_1$ respectively, indicated by colored arrows), specialization dynamics closely follows the inputs' dynamics. Networks showed with 1 (dark blue), 10 (teal) and 100 (yellow) active inter-module connections. Data are presented as mean values with a shaded standard error envelope.

but may not be the best measure of structural modularity (although we did not systematically investigate alternatives).

Structural modularity may well be one component of a richer approach to understanding functional specialization, including more complicated architectures, different learning rules, training regimes, datasets and neuron model (for example, a spiking neural network would have very different bandwidth to an equivalent artificial neural network). By systematically varying details of our toy network, we found that as a first step towards such a richer approach, introducing resource constraints was a reliable way to encourage stronger functional specialization. Intuitively this makes sense, as when computational resources are abundant there is no incentive to share or re-use these resources. Having shown that the link from structure to function is not as straightforward as one might think, an exciting follow-up research question would be to investigate the link between functional and compositional aspects of modularity. This link has already proved harder to understand than expected: even when neural networks do attain functionally specialized organization they do not always make compositional use of these modules[44,50,51].

To address the profoundly worrying climate emergency, in the light of the rapid growth in energy costs of AI, mechanisms that encourage parsimonious solutions are likely to be increasingly important. Neuromorphic engineering has the potential to take AI and robotics to a new era of scalable and low-power systems, away from the ever bigger and hungrier nature of current trends. Edge devices, equipped with neuromorphic sensors and processors would be a way for intelligent agents to integrate into society effectively and responsibly[52]. Such a decentralized, agent-centric perspective thus directly implies a modular organization, and seeing how such autonomous agents would be resource-constrained, the framework we developed would seem to be well fitted to better understanding their dynamics. Moreover, internal agents' dynamics would also be affected: neuromorphic chips function under the same kind of resource constraints as the brain does, with bandwidth and energy consumption being core preoccupations. Neuromorphic hardware specifically designed to implement networks with small-world topologies have also recently been put forward[53]. Finally, agents equipped with multiple neuromorphic sensors would naturally be suited to a modular or hierarchical organization, composed of unisensory and multisensory areas. The issues discussed in this paper may then prove important as we enter a new era of intelligent neuromorphic autonomous agents.

Our approach has been very handcrafted, with precise control over structural modularity and connectivity in the networks. However, other studies have taken a more emergent approach: various researchers[54,55] have investigated whether modular properties can emerge directly from the first principle of minimizing connection costs. Spatially-embedded networks, regularized to minimize connection costs while learning, do end up displaying modular and small-world features, but we note that both works had to introduce an additional regularization or optimization technique to see it emerge. Understanding if, and if so how, structural and functional modularity can emerge from purely low level and naturalistic principles outside of a controlled setup thus remains an open question. The search for such low level principles as a factor driving emergence of higher level concepts is nonetheless very promising. Another recent study shows how regularizing for energy efficiency could lead networks to organize into a predictive coding regime[56], and is a good example of how taking into account physical substrates and constraints can lead to a better understanding of networks' function.

We investigated relatively small networks, solving simple tasks. This enabled us to separate the multiple factors influencing modularity, and to thoroughly explore the parameter, architecture, and environment space. What's more, the simplicity of our setup makes for an ideal proving ground for concepts of modularity: if the current static definition of specialization already shows its limits here, it would

then seem too simple to describe complex and dynamic real-world systems. However, the opposite argument could also be made, suggesting that such a simple setup prevents us from seeing the emergence of functional specialization. Other works have taken more holistic approaches, studying both structural and functional modularity in trained networks at the same time[19,57,58]. Such an approach could reveal even further de-correlation of function and structure, as in refs. [19,58] where function is shown to emerge even without modular structure. In that sense by focusing only our efforts, and the scope of our metrics, on the two structurally defined modules we could already be missing functionally specialized yet decentralized structures. Further extensions of this work could also take into account multiple modules and tasks, and introduce varying environments. Indeed, such environments have been shown to favor modular solutions[59,60], and could also lead to different time-scales in specialization dynamics (lifelong learning). Not only that, but it has been shown that multi-task learning can drive the emergence of modularity[61,62], and as such would be interesting to investigate. In that context, the concept of systematic generalization is of crucial importance, and the ability for an agent to recombine and reuse previously acquired knowledge could prove decisive to its survival. Such an ability being better enabled through modular architectures[50,63,64], we can see how the necessity for generalization could also promote modular architectures to emerge (although this has yet to be shown, as discussed above). Finally, we only looked at the limited learning rule of supervised learning with backpropagation, and other learning methods could potentially lead to different results, as suggested by[65].

When it comes to the brain, although the concepts of structural and functional modularity are useful and well established, researchers are increasingly studying what this means in a networked, dynamic, and entangled way[4,5]. The brain is a messy, complicated system and we may not find clean and clear-cut concepts of modularity to apply directly. However, using artificial neural networks does give us a clean and controlled setup to explore these concepts. We can subsequently take these methods and apply them in more complex, uncontrolled environments, including artificial systems working with real world data, or biological systems. For example, a combined approach using artificial neural network decoders of electrophysiological recordings could enable us to measure specialization with the definitions introduced here. We conclude that using ANNs as a testing ground to develop more rigorous conceptions of modularity which can then be applied to experimental data could be a very fruitful approach for both neuroscience and machine intelligence.

## Methods

The goal of our work is to investigate the relationship between structural and functional modularity. To that end, we need to design flexible models, environments, and tasks, upon which we have precise control, to study this relationship in a controlled manner. We also need to understand and measure specialization and to do so we devise metrics that allow us to quantify this elusive concept.

### Environment: data and tasks

Conceptually, we think of an environment as composed of underlying variables, some of which are crucial to the task at hand (which we will call *decision variables*), while the remainder are needed only to fully reconstruct the exact observation but can be discarded when performing the task (*irrelevant variables*). Put together, those variables fully specify the environment. However, in carrying out a task we are only interested in capturing some high-level features of the world: we therefore define global tasks to be computations on *decision variables*.

In this framework, recovering the *decision variables* from a stimulus is what constitutes the **sub-tasks**: the essential building blocks which can then be used to perform a more general **global-task**. The precise form of stimulus can vary, representing different sets of

*irrelevant variables*. Changing the nature and structure of this stimulus is what allows us to control how difficult it is to recover a decision variable, hence the difficulty of the **sub-tasks**. We use (1) pairs of written MNIST[66] digits (with each digit drawn from the same set), or (2) pairs of EMNIST letters[67] (with each letter drawn from separated sub-sets). The **global task** is the precise computation to be performed after decoding the **sub-tasks**, and can also vary, allowing for control over the amount of cooperation needed by the agents. We train the networks on a parity-based choice:

$$\mathcal{T}_\sigma = \mathcal{D}_1 \cdot \sigma + \mathcal{D}_2 \cdot (1 - \sigma) \text{ where } \sigma = (\mathcal{D}_1 + \mathcal{D}_2)\%2 \qquad (1)$$

$\mathcal{D}_1$ and $\mathcal{D}_2$ are the numerical values of both digits (or the indices of the letters in the case of E-MNIST), and $\sigma$ is the parity of the sum, (ie $\sigma = 0$ when both are the same parity, and $\sigma = 1$ otherwise, with % being the modulo operator in Eq. 1). This task requires the network as a whole to classify both $\mathcal{D}_1$ and $\mathcal{D}_2$, but if done in a modular way each module can recognize only one digit and communicate only a single bit to the other: the parity of its own digit. This allows us to examine networks at extreme levels of sparse communication that can still solve the task. Let's take a look at the example in Fig. 1A to make the task clearer:

1. Two random digits $\mathcal{D}_1$ and $\mathcal{D}_2$ are sampled. In this case, $\mathcal{D}_1 = 0$ and $\mathcal{D}_2 = 3$.
2. Digits are fed to the network through their input weights (either all-to-all or one-to-one), at every time-step.
3. Modules then need to determine if digits are the same parity ($\sigma = 0$) or not ($\sigma = 1$). In this case, $\sigma = 1$.
4. In the scenario where $\sigma = 0$, the network must output the prediction $\mathcal{D}_1$. If not, the network must output $\mathcal{D}_2$. In this case, the correct label to this particular sample is $\mathcal{T}_\sigma = \mathcal{D}_2 = 3$.

Note that the task is unbalanced, since the case $\mathcal{D}_1 = \mathcal{D}_2$ provides information on both digits, making $\mathcal{D}_2 = \mathcal{T}_\sigma$ more frequent than $\mathcal{D}_1 = \mathcal{T}_\sigma$. To balance this, we train our networks to predict both:

$$\mathcal{T}_\sigma = \begin{cases} \mathcal{T}'_\sigma = \mathcal{D}_1 \cdot \sigma + \mathcal{D}_2 \cdot (1 - \sigma) \\ \mathcal{T}''_\sigma = \mathcal{D}_1 \cdot (1 - \sigma) + \mathcal{D}_2 \cdot \sigma \end{cases} \qquad (2)$$

The models consist of recurrent neural networks to allow for communication between modules, and we run them for a number of time steps (between 2 and 5). In most cases, stimuli are presented identically at every time step, but in "Function specialization can vary dynamically" they are switched on at a random time step and then remain on (Supplements 1.5.2). In that section we also add independent noise to the inputs at each time step (see details in Supplements 1.5.1).

## Networks

**Architecture**: We use a family of model architectures, composed of pairs of modules (single-layer recurrent neural networks of $n$ neurons) that are densely connected internally and sparsely connected between modules (a fixed fraction $p$ of the possible synapses, leading to a number of synapses $p_s$). To be precise, these modules consist of vanilla RNNs, but the code is written to allow the use of other modules types, such as GRUs. The modules can either have separate or shared inputs, i.e. either each module only receives input from one digit image, or each module receives input from both digit images. Similarly, they can have either separate or shared outputs (pathway structure). See below for how the network decision is made in these two cases. We can either connect the module directly to its output or include a narrow bottle-neck layer to restrict the bandwidth before readout. We vary $n$, $p$, the pathway structure, and the presence of the bottleneck layer. A typical network architecture is shown in Fig. 1A, and all possible global architectures are shown in a minimal fashion in Fig. 1B.

The choice of a recurrent rather than feed-forward architecture was made to keep a consistent architecture throughout the paper and to simplify the definitions of functional specialization and modularity.

**Decision**: Modules produce logits as their output, later used to compute the global network decision. As we've just seen, modules can either feed into a shared readout, (in which case the readout directly provides the network's decision), or have their own separate readouts. In the latter case, modules compete for decision via a winner-take-all scheme, to ensure that no communications is happening at the readout-layer level. In the separate readout case, each module is densely connected to a readout layer $\mathbf{r}^{(m)}$. Given the two readout layers, we compute the actual decision of the global model using a *max* decision, explained in the following paragraph:

$$\mathbf{r}^{out} = \mathbf{r}^{(\mu)} \text{ where } \mu = \arg\max_{m \in 0, 1} \max \mathbf{r}^{(m)}. \qquad (3)$$

For each input sample, the module with the highest overall "certainty" (largest value) takes the decision. More precisely, the global output layer of the network $\mathbf{r}^{out}$ is defined to be the output layer of the sub-network with the largest maximum value. In practice, that means that only the module having produced the largest overall value is being used in computing the resulting loss value, and that gradients are back-propagated to the other (non-deciding) module only via the (sparse) communication weights. This choice was made to minimize interactions between the modules other than the explicit sparse connectivity, ensuring the necessity of training the sparse inter-module weights.

## Structural modularity

We define the fraction of connections between two modules of size $n$ as $p \in [1/n^2, 1]$. The same fraction of connections is used in each direction. The smallest value is $p = 1/n^2$ corresponding to a single connection in each direction, and the largest fraction $p = 1$ corresponds to $n^2$ connections in each direction (all-to-all connectivity).

We adapt the $Q$ metric for structural modularity[16] to a directed graph:

$$Q = \frac{1}{M} \sum_{ij} (A_{ij} - P_{ij}) \delta_{g_i g_j}, \qquad (4)$$

where $M$ is the total number of edges in the network, $A$ is the adjacency matrix, $\delta$ is the Kronecker delta, $g_i$ is the group index of node $i$ (which module the node is in), and $P_{ij}$ is the probability of a connection between nodes $i$ and $j$ if we randomized the edges respecting node degrees. For our network, we can analytically compute (Supplements 1.1):

$$Q = \frac{1}{2} \cdot \frac{1-p}{1+p}. \qquad (5)$$

This varies from $Q = 0$ when $p = 1$ (all nodes connected, so no modularity) to $Q = 1/2$ for $p = 0$ (no connections between the sub-networks, so perfect modularity). Note that for $g$ equally sized groups in a network, the maximum value that $Q$ can attain is $1 - 1/g$.

## Metrics

In order to compute a metric of functional modularity at the network level, we start by constructing a measure of the strength of the relationship between the activity of a module $m$ and a sub-task (predicting digit $k$). We use three such measures (described below) based on performance or correlation. We write this as $\mathcal{M}(m, k) \in [b, 1]$ where a value of $b$ means no link (no performance or correlation) and a value of 1 means a strong link (perfect performance or correlation). $b$ represents the base value of a metric, and indicates a null relationship between an agent and a sub-task. We use this (metric-specific) value to normalize properly each metric, resulting in the normalized

module-and-digit specific measure:

$$\bar{\mathcal{M}}(m,k) = \frac{\mathcal{M}(m,k) - b}{1 - b} \in [0,1] \qquad (6)$$

From this measure that is specific to a particular module and digit $(m, k)$ we construct a module-specific measure of specialization by computing the difference

$$\mathcal{F}^m = \bar{\mathcal{M}}(m,0) - \bar{\mathcal{M}}(m,1) \in [-1,1]. \qquad (7)$$

This measures how much more linked the module $m$ is to one digit rather than the other. This has value 1 if the module is fully specialized on digit 0, or -1 if it is fully specialized on digit 1. Finally, the overall specialization of the network is maximum if both modules are fully specialized on opposite digits:

$$\mathcal{F} = \frac{|\mathcal{F}^0 - \mathcal{F}^1|}{2} \in [0,1] \qquad (8)$$

With this general structure in place, we define the three core measures as follows:

$\mathcal{M}_{pr}$: **Module probing** (Fig. 1C.1). After training on the global task above, we freeze the model and re-train a separate neural network to classify one of the two digits from the outputs of only one of the modules $m$. With this measure, the module $m$ is specialized on digit $k$ if the accuracy of classifying that digit $\mathcal{M}_{pr}(m,k)$ is high, and the accuracy of classifying the other digit is low. This corresponds to property 1 from the introduction and is also similar in spirit to the information bottleneck principle[68] in measuring how much irrelevant information has been discarded. The base value $b$ is chance accuracy (0.1 when computing MNIST digits).

$\mathcal{M}_{ab}$: **Module ablation** (Fig. 1C.2). Following the concept of separate modifiability (property 2), we once again freeze the network after training on the global task, but this time use a readout network common to both modules to predict the digits. The ablations metric measures the loss of performance when masking an entire agent's state on a given sub-task. We say the module $m$ is specialized on digit $k$ if its ablation critically impairs classification on this digit but not the other. The base value $b$ is once again chance accuracy.

$\mathcal{M}_{cr}$: **Hidden state correlations** (Fig. 1C.3). Finally, following the concept of domain specificity (property 3) we aim to understand the input-state relationship of modules of the network. Specifically, we measure how changing one decision variable (digit) while holding the other fixed impacts each module's state. We say the module $m$ is specialized on digit $k$ if there is a higher Pearson correlation coefficient between hidden states resulting from pairs of examples $x$ and $x'$ where digit $k$ of the two examples is the same, compared to the base correlation. This base correlation is the mean correlations of hidden states across all digit pairs (not holding one fixed), and is used as the normalizing base value $b$. Recently, work examining the representational similarities of neural nets has shown that because of the inherent non-linearity present in ANNs, a better suited metric to compare multiple networks, or layers of a networks, is the CKA measure[69]. While we agree that this is important when comparing modules that could have the same function with different parametrisations, we are only interested here in measuring the self-similarity of the modules, thus a standard correlation analysis is sufficient.

## Reporting summary
Further information on research design is available in the Nature Portfolio Reporting Summary linked to this article.

## Data availability
We provide all the data needed to reproduce the figures shown in this manuscript, and provide the code to reproduce said data in https://doi.org/10.6084/m9.figshare.27161427. It is to be noted that this has been a long-running project with some experiments needing substantial computational resources ("Environmental structure and resource constraints determine specialization" especially). To this end, we would encourage interested readers to investigate the data available rather than trying to re-run the entire parameter search.

## Code availability
The original code used to generate experiments and results from this work is available at https://github.com/GabrielBena/community-of-agents/tree/paper_version. However, we recommend interested readers to consult this simplified repository, which is cleaner, faster and contains everything needed to reproduce all results shown in this paper: https://github.com/GabrielBena/specialization-dynamics.

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

## Acknowledgements
We thank the team of the Neural Reckoning lab, especially Marcus Ghosh and colleagues at Imperial such as Maxence Faldor and Jeremy Pitt for fruitful insights, feedback and discussions.

## Author contributions
Both authors contributed equally to the work. G.B. and D.G. conceptualized the ideas, methodology and goals. G.B contributed most of the code and ran experiments. G.B. and D.G. wrote the manuscript.

## Competing interests
The authors declare no competing interests.
