## [Transparent Peer Review file · Nature Communications]

Dynamics of specialization in neural modules under resource constraints

Corresponding Author: Dr Dan Goodman

Version 0:

Reviewer comments:

Reviewer #1

(Remarks to the Author)

The paper by Bena and Goodman addresses a question of fundamental importance in neuroscience. The question of brain modularity has always been central to the discipline itself, though very rarely addressed head-on in a formal manner. The investigation is very timely and the study provides important contributions to the field. I think this paper is really important! I have a few suggestions having in mind enhancing the impact of the study.

While the questions addressed are well motivated, some of the text is quite hard to follow and could use careful editing to make it more broadly accessible. This includes relatively minor issues, such as explaining, for example, parity-based choice and single parity bit, to central components of the paper.

-- I struggled with both Figure 3 and 4. Please try to explain it more clearly.

-- Subsection "Metabolic cost constraints": was it supposed to link to any figure?

-- Subsection "Bottleneck": cooperation is suggested to be associated with modularity and I think I understand it, but could be thought in terms of decreasing modularity as it involves interactions.

-- Section "Output scheme" was pretty hard and some parts felt counterintuitive. It really points to the challenges of guessing modularity from architectural properties, which the paper shows convincingly it's really challenging! It's also challenging to compare the panels because many of the color scales are different. Isn't that a problem??

-- I understood some of the results in Fig. 4 but not nearly enough, really hard!

-- The paper is yours to write, so I hate to say this but the Discussion might need a major rewrite and adoption of a more traditional style to highlight what was accomplished and the merits of the study.

Also, the authors might like to check some great work on the logic of double dissociations and how they can fail: Young, M. P., Hilgetag, C. C., & Scannell, J. W. (2000). On imputing function to structure from the behavioural effects of brain lesions. *Philosophical Transactions of the Royal Society of London. Series B: Biological Sciences*, 355(1393), 147-161.

Reviewer #2

(Remarks to the Author)

I enjoyed reading the submitted work. It was well written, well-structured, well presented and well thought. I congratulate the authors for putting together such an interesting piece and successfully conveying the main findings. This work provides compelling evidence for the dissociation between the structural modularity and the functional one. It, therefore, has major implications for both neuroscientists aiming at understanding cognition from the brain's intricate network structure, and for AI researchers aiming to incorporate biologically plausible architectures. The work is not the first one addressing this broad issue but (to my knowledge) is novel in addressing this specific relationship between structural and functional modularity.

Below, I list a few minor comments and suggestions that can (hopefully) improve the readability and robustness of the work:

1. Authors mention the limitations of the single-site lesions, I know some works that address exactly this controversial issue and can be cited to provide evidence for this sentence. I encourage authors to take a look and choose one or two:

- Fakhar, K., & Hilgetag, C. C. (2022). Systematic perturbation of an artificial neural network: A step towards quantifying causal contributions in the brain. *PLoS Computational Biology*, 18(6), e1010250. <https://doi.org/10.1371/journal.pcbi.1010250>

- Young, M. P., Hilgetag, C. C., & Scannell, J. W. (2000). On imputing function to structure from the behavioural effects of brain lesions. *Philosophical Transactions of the Royal Society of London. Series B, Biological Sciences*, 355(1393), 147–161. <https://doi.org/10.1098/rstb.2000.0555>

- Young, M. P., Hilgetag, C., & Scannell, J. W. (1999). Models of paradoxical lesion effects and rules of inference for imputing function to structure in the brain. *Neurocomputing*, 26-27, 933–938. [https://doi.org/10.1016/S0925-2312\(99\)00012-0](https://doi.org/10.1016/S0925-2312(99)00012-0)

2. Authors mention "credit assignment" as a problem that both ANNs and BNNs (Biological Neuronal Networks) should solve. I suggest them to expand this with a sentence or two, explaining the concept for the naive audience.

3. The last line of the same paragraph about studying modularity of ANNs can also be backed up by one or two of the previous works:

- Marton, C. D., Lajoie, G., & Rajan, K. (2021). Efficient and robust multi-task learning in the brain with modular task primitives. In *arXiv [cs.AI]*. arXiv. <http://arxiv.org/abs/2105.14108>

- Clune, J., Mouret, J.-B., & Lipson, H. (2013). The evolutionary origins of modularity. *Proceedings. Biological Sciences / The Royal Society*, 280(1755), 20122863. <https://doi.org/10.1098/rspb.2012.2863>

4. I encourage the authors to make the task crystal clear for the non-expert readers since this work is interesting for neuroscientists from a wide range of backgrounds. Although it is explained using simple equations, I would add a paragraph going through each step, mentioning the difference between MNIST and EMNIST, maybe even add a separate figure for it. Understanding the task is one of the key parts of the work and I recommend authors to show it to even show this part to neuroscientists working in the cognitive/behavioral fields to confirm it is clear for people with a background in biology or psychology.

5. Following on that, I recommend authors to elaborate the figures a bit more in their captions. For instance, Figure 1.C gives the impression that one module is fixed, and the other is being retrained while in the text it is mentioned that a separate network is being re-trained.

6. I see "isn't, we've, wouldn't, ..." in the text, personally, I support having a friendlier tone in scientific literature, but the convention requires a rather formal approach. I leave the decision to change/keep the text to the authors and the editor.

7. I understand the reasoning behind "smaller number of neurons/sparser network to represent metabolic cost" but I think this won't convince many others. For instance, in neuromorphic computers, the activity of each unit is as important (if not more) so I wonder if the authors can either discuss this point or add an extra analysis comparing the network wide "energy consumption" of architectures. I expect to see a compensation in sparser networks by having more active units. If not, then the authors have their point supported. A handy metric here is to simply sum the L2 norms (energy) of the units (`numpy.linalg.norm` does the trick.) Another approach has been proposed by the paper below:

Ali, A., Ahmad, N., de Groot, E., Johannes van Gerven, M. A., & Kietzmann, T. C. (2022). Predictive coding is a consequence of energy efficiency in recurrent neural networks. *Patterns (New York, N.Y.)*, 3(12), 100639. <https://doi.org/10.1016/j.patter.2022.100639>

8. Figure 4 is one of the figures that can benefit from more explanation. For example, it is not clear what the vertical dashed lines represent. Maybe even a color bar to have a rough idea of which p corresponds to which color. But that is not really necessary since I believe the trend is more important than the actual values.

9. The authors mention the often neglected dynamics of modularity in the second line of the discussion. I agree with their statement, but I think, for completeness, they should mention that there are works on this topic and the methods are there in case the reader wanted to follow them up. Here are two seminal examples:

Shine, J. M., Bissett, P. G., Bell, P. T., Koyejo, O., Balsters, J. H., Gorgolewski, K. J., Moodie, C. A., & Poldrack, R. A. (2016). The Dynamics of Functional Brain Networks: Integrated Network States during Cognitive Task Performance. *Neuron*, 92(2), 544–554. <https://doi.org/10.1016/j.neuron.2016.09.018>

Bassett, D. S., Wymbs, N. F., Porter, M. A., Mucha, P. J., Carlson, J. M., & Grafton, S. T. (2011). Dynamic reconfiguration of human brain networks during learning. *Proceedings of the National Academy of Sciences of the United States of America*, 108(18), 7641–7646. <https://doi.org/10.1073/pnas.1018985108>

10. I wonder if the authors explored the excitatory/inhibitory ratio of the inter-module connections? Based on the previous

works on circuit mechanisms of perceptual decision-making, I expect them to be mostly inhibitory, at least in the case with separate outputs. Also, I think with increasing sparsity, the network should prioritize inhibition between modules so the ratio should get to larger values. The authors are welcome to conduct the analysis, but I don't see it being essential for their story, so they can also speculate this in the discussion if that's too much to do.

11. Another possible extension is to add more modules than two. E.g., what happens if there are three modules? Will the third module divide into two or the allocation would be asymmetric given some feature in the input? Would 5 modules just make the network more robust by providing a backbone for redundant computation? Again, not essential but I'm curious to at least have the authors' thoughts on this. I feel these two points (10 and 11) can potentially expand into a separate paper, exploring the dynamics of interaction and resource allocation in modular neural networks.

I wish the authors the best of luck in their future works and again, well done on this interesting project.

Reviewer #3

(Remarks to the Author)

This paper studies the factors leading to the emergence of functional modularity in artificial neural networks. Its main finding is that structural modularity is not a sufficient factor (except at extreme values). Hence, the Authors conclude that, similarly, functional modularity in animal brains may not hinge entirely upon the brain structure.

This paper poses an exciting question and provides a solid motivation for its experiments. Its findings are non-trivial and help shed light on the very notion of modularity. Nevertheless, this paper also suffers from a series of limitations. Its conclusions are too broad as they are based on a single task (parity-based digit classification) and a single neural architecture (recurrent neural networks). At the same time, the formal description of such neural architecture is not fully satisfactory as it would not be possible to reproduce it based on the paper alone. Finally, the paper does not reference a large portion of the literature on this topic from the field of machine learning, which provides evidence that multi-task learning and dynamic, varying environments are the key to functional specialisation.

In light of both the strengths and weaknesses highlighted above, I recommend that this paper is accepted only after major revisions to address my comments below (divided by section).

****Abstract****

The abstract contains a conclusion (1) that does not seem to be warranted given the experiments reported in the paper, namely that "specialization can only emerge in environments where features of that environment are meaningfully separable". All the inputs consist of separable variables (two digits) so no control experiments are present.

The phrasing of the conclusion (3) in the abstract also seems too general: "findings are qualitatively similar across different network architectures" makes it sound like different neural architectures were tested (such as MLPs, CNNs, Transformers, etc.) whereas only (variants) of recurrent neural networks are included in the paper.

****Introduction****

Here the Authors make a compelling case for using in-vitro analyses of modularity based on artificial neural networks, given that biological neural networks can only be studied imperfectly through lesions or brain imaging. As a person expert in machine learning but not in neurosciences, I found this section easily digestible by a wide array of potential readers.

The Achilles' heel of this section is its lack of references, especially from the field of AI and machine learning, of previous work that attempted similar analyses on artificial neural networks. Some of these works employed tools similar to this paper's experiments. For instance, Meyes (2020) conducted ablation analyses on subpopulations of neurons.

Other missing references highlight some of the limitations of the present work, which I discuss below.

On measuring structural modularity Structural modularity was found to emerge even in fully connected neural networks (Watanabe 2019, Casper et al. 2022). In practice, when a connection is present but its weight is approximately 0, it becomes equivalent to a missing edge in the network. Hence, measuring the level of structural modularity based on the Q-metric, which ultimately depends on the ratio of cross-module connections p , may be insufficient.

On the variables leading to the emergence of functional modularity The emergence of functional specialisation in modules is known to be driven by multi-task learning (Yang et al. 2019, Dobs et al. 2022) and dropout regularisation (Lange et al. 2022), among others. This illustrates a fundamental limitation of this work, which focuses on a single (global) task. In fact, modular behaviour may emerge under different structural conditions when multiple global tasks, composed of different sub-tasks, are taken into consideration.

On defining compositional generalisation as a criterion for modularity Fodor's first principle hints at a definition of modularity as the ability to *re*compose modules in new combinations, which leads to systematic generalisation in new global tasks composed of known sub-tasks. For instance, Csordas et al. (2021) showed that this is not the case in vanilla artificial networks: modules corresponding to different sub-tasks are not re-used for similar sub-tasks and are not composed for global tasks requiring new combinations of sub-tasks. The present paper should consider discussing functional modularity from a compositional perspective, considering systematic generalisation as one of the analysis tools. Mittal et al.

(2022), for instance, showed how structural modularity through routing achieves better generalisation.

****Methods****

The formal description of the neural architecture is lacking as it does not provide several details necessary to reproduce it. For instance, the authors mention that it consists of a “recurrent neural network”. However, it does not specify: 1) what kind (a vanilla RNN, a GRU, an LSTM)? 2) how deep (i.e., how many layers)?

Moreover, it is not clear from the description how connections between modules operate. Are these connections from some dimensions of the input of layer l of module A to the output of layer l of module B? This should be clarified as this is a crucial component of the experimental setup.

The very choice of RNNs as neural architectures is dubious: since the data (pairs of digits) are not sequential, why not use an MLP or a CNN instead? Except for the experiments in §3.3, where information is gradually unveiled, the input is identical for all time steps in the current setup.

The decision process for separate readouts is also unclear. What is the output of the readout modules? There are glimpses of information ($d_{\text{out}} = \text{argmax } r_{\text{out}}$) that make it sound like it is a probability distribution over a categorical variable (parity-based choice). But then what does it mean that r_{out} is chosen via a max function? Are you picking the module with the highest probability mass placed on any digit? Again, the lack of clarity hinders a potential reader from fully appreciating your experimental setup.

The subsection on metrics was well written and all the three considered metrics are well motivated in light of Fodor’s criteria for modularity. What I found confusing was the name of the first metric, “Bottleneck retraining”. If you are freezing and then training a new, separate network (as common in probing techniques) then you are not *re*training. Also, it is not properly a bottleneck as it directly predicts the output rather than constraining information in intermediate layers. Consider renaming this metric to “probing of the trained modules” or similar.

Another possible criticism of the third metric (“hidden state correlation”) is that non-linear neural networks may encode the same function through different parameterisations (which also results in different hidden states). In other words, functional similarity does not necessarily imply parametric similarity. Consider at least measuring linear CKA in lieu of Pearson’s correlation as it captures similarity invariant to orthogonal transformations.

****Results****

Overall, this section has an in-depth discussion of the different factors that may affect modularity, which is stimulating.

One limitation is the discussion of architecture choices and in particular output schemes. If anything, the bottleneck variant where modules compete for readout creates an inductive bias *against* modularity as defined in this task, because modules would tend to specialise towards subsets of *examples* rather than different subsets of variables within each example.

****References****

Stephen Casper, Shlomi Hod, Daniel Filan, Cody Wild, Andrew Critch, and Stuart Russell. Graphical clusterability and local specialization in deep neural networks. In ICLR 2022 Workshop on PAIR2Struct, 2022.

Katharina Dobs, Julio Martinez, Alexander JE Kell, and Nancy Kanwisher. Brain-like functional specialization emerges spontaneously in deep neural networks. *Science Advances*, 8(11):eabl8913, 2022.

Lange, R. D., Rolnick, D. S., & Kording, K. P. (2022). Clustering units in neural networks: upstream vs downstream information. arXiv preprint arXiv:2203.11815.

Meyes, R., de Puiseau, C. W., Posada-Moreno, A., & Meisen, T. (2020). Under the hood of neural networks: Characterizing learned representations by functional neuron populations and network ablations. arXiv preprint arXiv:2004.01254.

Mittal, S., Bengio, Y., & Lajoie, G. (2022). Is a modular architecture enough? *Advances in Neural Information Processing Systems*, 35, 28747-28760.

Chihiro Watanabe. Interpreting layered neural networks via hierarchical modular representation. In *Neural Information Processing - 26th International Conference, ICONIP 2019, Sydney, NSW, Australia, December 12-15, 2019*, pp. 376–388.

Guangyu Robert Yang, Madhura R. Joglekar, H. Francis Song, William T. Newsome, and Xiao-Jing Wang. Task representations in neural networks trained to perform many cognitive tasks. *Nature Neuroscience*, 22(2):297–306, 2017.

Reviewer #4

(Remarks to the Author)

Thank you for this very interesting paper.

Please find a formatted version of the review in the attached pdf.

Review of “Dynamics of specialization in neural modules under resource constraints”

Based on experiments with a modular (2 pathway architecture in 3 different configurations) the paper makes the following claims

1. Structural modularity isn't sufficient to lead to functional specialization unless there are extreme resource constraints.
2. Correlated inputs lead to correlated states (i.e., disfavor specialization)
3. Temporal dynamics of specialization are relevant and should be studied more.

I agree that modularity is an extremely important topic for the neuromorphic field as the connectivity is constrained to a topology with strong local, but sparse global connectivity, like a small world topology. In addition, it looks like the topic of modularity is understudied, especially from a neuromorphic perspective.

Unfortunately, it is not clear how the conclusions of this paper are meaningful for neuroscience or neuromorphic engineering even though this seems to be the motivation for the paper. The discussion on the relevance for neuromorphic approaches seems to be an appendage and rather vague.

Additionally, some claims are a bit questionable. Claim 1. seems to depend strongly on the specific choice of task, architecture, and definition of modularity. Claim 2. seems almost trivial given the way specialization is measured here and claim 3 is vague.

Nevertheless, I appreciate the originality and simplicity of the approach as it can lead to direct and easy to understand conclusions on a very complex topic (as long as we are aware of the risks of such simple conclusions).

Additionally, one cannot expect a single paper to be comprehensive on such a complex topic, but a paper that aspires to be published in such a renowned journal and makes broad claims should try to address some of these issues (see more detailed info below).

So, if revised accordingly and significantly, this paper might serve as a starting point for a deeper study of modularity from a neuromorphic perspective.

Do the conclusions of this paper generalize to more complex architectures?

The topic of modularity is very complex, as there are many forms of modularity and specialization (fused/hybrid networks, clusters within networks, parallel pathways, sequential blocks, modularity within nodes, etc., see Amer and Maul 2019). As far as I can tell, all types of modularity can also be observed in the brain on different levels, but only one definition is investigated here.

This already indicates that the simple conclusions that are drawn in this paper are only limited to a specific case and might not have the general relevance that is implicitly claimed.

The most important limitation in my opinion is that the paper only looks at modularity related to the input and the output of the network, but the more interesting and important modularity arises due to the task and the structure of the hidden layers.

Especially modern neural network architectures are quite modular by design. For instance, transformers have a multi-head structure, and each head contains specialized parts.

Particularly, the paper claims that structural modularity is not sufficient to guarantee functional specialization, unless there are extreme constraints. The multi-headed transformer architecture seems to prove this wrong. In transformers, the multi-headed structure leads to functional specialization via the attention mechanism (which I would consider part of the structure) without extreme resource constraints.

Of course, one could argue that there are constraints in the number of neurons per head or the number of heads, which are relevant due to the huge size of the input and output spaces and the complexity of the task.

It is however also possible that, on the contrary, mechanisms like attention may be needed to make use of modularity and the conclusions in this paper simply cannot be applied to such architectures. I am not sure if this question can be answered based on the currently available evidence.

A number of papers (that are ignored in this work) show that neural networks are in fact quite modular (even though they cannot make good use of this modularity). For instance:

Csordás et al. 2021 provides a compelling way to measure modularity based on the weights and <https://arxiv.org/pdf/2010.02066.pdf>

Filan et al. 2021, Clusterability in neural networks <https://arxiv.org/pdf/2103.03386.pdf>

Hod et al. 2022, Quantifying local specialization in deep neural networks <https://arxiv.org/pdf/2110.08058.pdf>

Zhang et al. 2023, Emergent Modularity in Pre-trained Transformers

So, it is unclear if the claims of this paper are limited to the chosen simple toy architecture. This should be discussed or addressed by additional experiments.

An even more interesting aspect of modularity, that is essentially ignored in this paper (and also implicitly questions the paper's claims) is the effect of modularity on generalization. Here is a small selection of papers that address this topic:

Goyal et al. 2021, Recurrent independent mechanisms, <https://arxiv.org/pdf/1909.10893.pdf>

Kirsch et al. 2018, Modular networks: Learning to decompose neural computation <https://proceedings.neurips.cc/paper/2018/file/310ce61c90f3a46e340ee8257bc70e93-Paper.pdf>

Bahdanau et al. 2019, Systematic generalization: what is required and can it be learned? <https://arxiv.org/pdf/1811.12889>

While most of these papers construct modular models as an inductive bias, it seems plausible that if a task requires generalization capabilities, architectures could become modular through some form of optimization (be it evolutionary or

gradient-based). I.e., there are (structural and non-structural) drivers of (structural and functional) modularization beyond resource constraints. (An interesting aspect to investigate, of course, could be, if resource constraints and generalization could go hand in hand and constraints can lead to better generalization).

And here is yet another example where modularity seems to be useful due to task requirements:
Ellefsen et al. 2015, Modularity to avoid catastrophic forgetting, <https://doi.org/10.1371/journal.pcbi.1004128>

Are the conclusions of this paper relevant for neuroscience?

The previous criticism concerning the non-applicability to more complex architectures certainly applies to the brain. It is, in fact, fairly easy to come up with a counterexample to the claim that “structural modularity isn’t sufficient for functional specialization” (ok, I admit there is a chicken and egg problem here):

Efficient modulation in the brain by “spraying” a small region (structural module) with transmitters (e.g., dopamine) would fail if there was not also a certain extent of functional specialization.

I.e., if we assumed the structure as fixed, functional specialization would likely have to arise through optimization (so, for complex enough structures, structural modularity is (likely) sufficient for functional specialization).

Another example, that is closer to the idea of the paper might be the following: Let’s say we have a convolutional-like architecture that resembles visual cortex and an architecture that is designed to detect different temporal frequencies. Both start with random parameters and receive both visual and auditory input (let’s say spoken and written MNIST). Would we really assume, based on this paper, that the modules would not specialize on the respective suitable input given by the structure?

Additional comments

1. I assume the classification accuracy for the task is not reported as it does not play a role for modularity, but it seems relevant to at least mention it. Is it always 100%?

The presented networks are likely strongly overfitted to the task (there is no mention of a training, validation, or test set). This might have an influence on the given measures, as the level of training could affect modularity.

Especially in the extremely resource constrained cases, it matters if the networks are trained to 100% accuracy or with a fixed number of epochs (which might lead to lower accuracy). In the former case, I expect more specialization than in the latter. So, to be able to really understand and put the results in context, reporting the accuracy is needed.

2. “In this structural configuration, networks often resort to having a main decision-making module. Although the task is designed to be solvable in a modular fashion, having such a module take all decisions means it’s able to predict both digits and thus displays less specialization.”

This seems a bit contradictory. If there is a main decision-making module, it seems like there is a form of specialization, it is just not detected by the measure.

3. The paper makes some assumptions about the structure of how the networks solves the task that do not seem plausible or are at least not proven.

Section 2.2.1 talks about “sub-tasks” and a “global task”. This makes sense from a human perspective as that is how we would approach such tasks, but it is not shown that the network actually solves the task in such a manner. On the contrary, I think there is no reason to believe that. It will likely just find a shortcut. Instead of classifying digits separately, it can just learn directly which pixel combinations correspond to which output (in a nonlinear fashion, of course).

4. “We hypothesize that this is a result of the static nature of inputs, where all the information is readily available at every time step and so there is no advantage to ongoing communication after the initial burst.”

I do not fully agree with this interpretation. There is still “communication” between the networks as they are still connected. The other module will likely not create a memory of the opposite module as it can just read it out at each timestep. I.e., the specialization measure does not allow to judge communication. To show that one module develops a memory of what happens in the other, one would have to cut the connection in timestep 3 or 4 (and maybe run it longer).

I think that the fact that the modules have to create a memory of the opposite side (as they cannot rely on the input) may be the reason why one can observe overall lower specialization in the noisy input case.

5. Please also note the following recent paper that seems to be relevant:

<https://www.modulardeeplearning.com/>

<https://arxiv.org/pdf/2302.11529.pdf>

Minor comments to improve form, language, understandability, etc.:

1. The language is good, but for punctuation and to reduce informal and wordy language it might still help to use a grammar/language tool. An example for wordiness is: “such as audio-visual integration for example”

2. Typo: “and is run”

3. Equation numbering takes a break after (4)

4. Fig. 3: color bar should be labelled; The colormaps, at least in each lettered subfigure should have the same range. I would prefer all Figures to have the same range even if that makes it harder to see the small differences in subfigure C.

5. Fig. 4: Please label axes and add a colormap. The caption mentions the meaning, but it makes it much harder to read the figure. The colors are only explained in C. but mentioned in A. I would also not make the line dashed; this is confusing and unnecessary as the lines are clearly separable.

6. Typo: “and is run for a number of time steps”

7. Please define % (modulo) and D1, D2 (numerical value of digits?) in Eq.1.

8. I would suggest to first explain the task, then the networks, as it is easier to understand the network structure once it is clear what the input and output are.

9. The part about “Metabolic constraints” (bottom of page 7) is not explained well. Also, you should refer to Fig.3 in this

paragraph.

10. Please explain better what the outputs of the network are. Especially explain what the difference between separate and fused is.

11. "Functional specialization is usually understood as a static property" Please provide a reference for this statement.

Version 1:

Reviewer comments:

Reviewer #1

(Remarks to the Author)

The authors have addressed my prior concerns/suggestions.

Reviewer #2

(Remarks to the Author)

I'm happy that the authors found my comments instructive and implemented them fully. I have no further suggestions at this point, of course, one can always continue with "what if" and "how about that"s, but given the objective of this study, I believe the work is mature enough as is. I'm looking forward to reading further studies that follow this one, particularly with more complex tasks and more modules.

I wish the authors the best of luck.

Reviewer #4

(Remarks to the Author)

Please find my detailed comments in the attachment.

General remarks

We would like to thank the reviewers for their careful and detailed reading of our work, and for their comments that have enabled us to greatly improve the clarity of the paper throughout, write a much improved discussion, as well as increasing the breadth of relevant literature cited and discussed.

Below, we have replied to each point with the original review in black and our response in blue. We have included in our resubmission a clean copy of the new paper as well as a diff showing additions in blue and deletions in footnotes in red. Unfortunately, the diff shows sections and paragraphs that have been moved as if they were entirely new, and we were unable to find a workaround for this.

Reviewer #1 (Remarks to the Author):

The paper by Bena and Goodman addresses a question of fundamental importance in neuroscience. The question of brain modularity has always been central to the discipline itself, though very rarely addressed head-on in a formal manner. The investigation is very timely and the study provides important contributions to the field. I think this paper is really important! I have a few suggestions having in mind enhancing the impact of the study.

While the questions addressed are well motivated, some of the text is quite hard to follow and could use careful editing to make it more broadly accessible. This includes relatively minor issues, such as explaining, for example, parity-based choice and single parity bit, to central components of the paper.

- I struggled with both Figure 3 and 4. Please try to explain it more clearly.
 - We agree that the figures were difficult to follow, and so we have modified them as follows:
 - Fig 3 now features consistent log-scale color bars across all 3 subplots. That makes it easier to compare the three architecture choices meaningfully, while still showing the pattern and relationships between parameters p , n and c
 - The results section for Fig. 3 has also been improved, and details added.
 - Fig. 4 was changed extensively: captions and labels make it clearer what to look for in the patterns showed, and we've added the missing colorbar and axis labels as well.
- Subsection "Metabolic cost constraints": was it supposed to link to any figure?
 - We added a reference to Fig. 3.
- Subsection "Bottleneck": cooperation is suggested to be associated with modularity and I think I understand it, but could be thought in terms of decreasing modularity as it involves interactions.
 - We agree the use of the word 'cooperation' is confusing here. We have reworded this section to make it clearer.
- Section "Output scheme" was pretty hard and some parts felt counterintuitive. It really points to the challenges of guessing modularity from architectural properties, which the paper shows convincingly it's really challenging! It's also challenging to compare the panels because many of the color scales are different. Isn't that a problem??

- We have rewritten the text to improve clarity. We agree that a shared color scale across panels is better. To make the patterns visible with this change, we switched to a log color scale cropped at a minimum specialization value of 0.01, and a different color map with higher dynamic range.
- I understood some of the results in Fig. 4 but not nearly enough, really hard!
 - We agree that overall Fig 4 can be hard to digest. We have added labels, texts and hopefully made it overall clearer, as well as improving the caption.
- The paper is yours to write, so I hate to say this but the Discussion might need a major rewrite and adoption of a more traditional style to highlight what was accomplished and the merits of the study.
 - We have done a fairly major rewrite of the Discussion section, both to improve clarity and to cover several points raised by other reviewers. The discussion also now starts with a more traditional highlight of what was accomplished.
- Also, the authors might like to check some great work on the logic of double dissociations and how they can fail: Young, M. P., Hilgetag, C. C., & Scannell, J. W. (2000). On imputing function to structure from the behavioural effects of brain lesions. *Philosophical Transactions of the Royal Society of London. Series B: Biological Sciences*, 355(1393), 147-161.
 - Thank you for this extremely relevant recommendation, we have added a citation in the introduction.

Reviewer #2 (Remarks to the Author):

I enjoyed reading the submitted work. It was well written, well-structured, well presented and well thought. I congratulate the authors for putting together such an interesting piece and successfully conveying the main findings. This work provides compelling evidence for the dissociation between the structural modularity and the functional one. It, therefore, has major implications for both neuroscientists aiming at understanding cognition from the brain's intricate network structure, and for AI researchers aiming to incorporate biologically plausible architectures. The work is not the first one addressing this broad issue but (to my knowledge) is novel in addressing this specific relationship between structural and functional modularity.

Below, I list a few minor comments and suggestions that can (hopefully) improve the readability and robustness of the work:

- 1. Authors mention the limitations of the single-site lesions, I know some works that address exactly this controversial issue and can be cited to provide evidence for this sentence. I encourage authors to take a look and choose one or two:
 - - Fakhar, K., & Hilgetag, C. C. (2022). Systematic perturbation of an artificial neural network: A step towards quantifying causal contributions in the brain. *PLoS Computational Biology*, 18(6), e1010250. <https://doi.org/10.1371/journal.pcbi.1010250>
 - - Young, M. P., Hilgetag, C. C., & Scannell, J. W. (2000). On imputing function to structure from the behavioural effects of brain lesions. *Philosophical Transactions of the Royal Society of London. Series B, Biological Sciences*, 355(1393), 147–161. <https://doi.org/10.1098/rstb.2000.0555>

- - Young, M. P., Hilgetag, C., & Scannell, J. W. (1999). Models of paradoxical lesion effects and rules of inference for imputing function to structure in the brain. *Neurocomputing*, 26-27, 933–938. [https://doi.org/10.1016/S0925-2312\(99\)00012-0](https://doi.org/10.1016/S0925-2312(99)00012-0)
 - Thank you for these very relevant recommendations, we have added citations in the introduction.
- Authors mention "credit assignment" as a problem that both ANNs and BNNs (Biological Neuronal Networks) should solve. I suggest them to expand this with a sentence or two, explaining the concept for the naive audience.
 - We agree and have added some text to this effect.
- The last line of the same paragraph about studying modularity of ANNs can also be backed up by one or two of the previous works:
 - - Marton, C. D., Lajoie, G., & Rajan, K. (2021). Efficient and robust multi-task learning in the brain with modular task primitives. In arXiv [cs.AI]. arXiv. <http://arxiv.org/abs/2105.1410>
 - [Clune, J., Mouret, J.-B., & Lipson, H. \(2013\). The evolutionary origins of modularity. *Proceedings. Biological Sciences / The Royal Society*, 280\(1755\), 20122863. https://doi.org/10.1098/rspb.2012.2863](https://doi.org/10.1098/rspb.2012.2863)
 - We added these citations.
- I encourage the authors to make the task crystal clear for the non-expert readers since this work is interesting for neuroscientists from a wide range of backgrounds. Although it is explained using simple equations, I would add a paragraph going through each step, mentioning the difference between MNIST and EMNIST, maybe even add a separate figure for it. Understanding the task is one of the key parts of the work and I recommend authors to show it to even show this part to neuroscientists working in the cognitive/behavioral fields to confirm it is clear for people with a background in biology or psychology.
 - Given how important the task is, we agree with the reviewer and have added an additional paragraph and figure.
- Following on that, I recommend authors to elaborate the figures a bit more in their captions. For instance, Figure 1.C gives the impression that one module is fixed, and the other is being retrained while in the text it is mentioned that a separate network is being re-trained.
 - One of the reviewers suggested the word 'probing' for what we called 'retraining' and this seemed clearer, so we have changed the text and Figure 1 to use this new word. We also modified the figure to make it clearer what is going on, and expanded the caption.
- I see "isn't, we've, wouldn't, ..." in the text, personally, I support having a friendlier tone in scientific literature, but the convention requires a rather formal approach. I leave the decision to change/keep the text to the authors and the editor.
 - We have changed most of these throughout the text while revising, although we do also prefer a friendlier tone and it was not an issue raised by the editor.
- I understand the reasoning behind "smaller number of neurons/sparser network to represent metabolic cost" but I think this won't convince many others. For instance, in neuromorphic computers, the activity of each unit is as important (if not more) so I wonder if the authors can either discuss this point or add an extra analysis comparing the network wide "energy consumption" of architectures. I expect to see a compensation in sparser networks by having more active units. If not, then the authors have their point supported. A

handy metric here is to simply sum the L2 norms (energy) of the units (numpy.linalg.norm does the trick.) Another approach has been proposed by the paper below:

- Ali, A., Ahmad, N., de Groot, E., Johannes van Gerven, M. A., & Kietzmann, T. C. (2022). Predictive coding is a consequence of energy efficiency in recurrent neural networks. *Patterns* (New York, N.Y.), 3(12), 100639. <https://doi.org/10.1016/j.patter.2022.100639>
 - Energy consumption could be decomposed into the cost of maintaining a cell regardless of its activity, plus the cost due to its activity. Our analysis only covers the first part which we feel is justified for the following reasons.
 - In biological systems, it has been shown that energy consumption of the brain is proportional to the number of neurons (Herculano-Houzel 2011). This may be because there is an upper limit to neuronal spike rates, and in addition there are spatial as well as energy constraints.
 - In neuromorphic systems, there is no maintenance cost per unit but there is a manufacturing cost. In general for low power devices, manufacturing costs have been estimated to outweigh lifetime energy costs (e.g. <https://bits-chips.nl/artikel/its-the-manufacturing-stupid/>).
 - In spiking neural networks, it's straightforward to measure activity cost as proportionate to number of spikes, however in artificial neural networks it is a tricky problem to model this and we are not convinced by any approach we have seen so far. We are not entirely sure what is meant by summing the L2 norms of the units – does it refer to activations, pre-activations or weights? We find the approach taken in Ali et al. (2022) interesting but not entirely satisfactory. It may be possible to address these issues with a suitable model, but this feels like a separate research problem: this would need to be included as an active regularisation term during training, where we would likely have to make a number of choices for which there is no natural choice (e.g. in Ali et al. 2022, they arbitrarily choose to weight the contributions of activity and synaptic transmission with factors 1/3 and 2/3), and would therefore add a lot of additional complexity to the model and make the results even harder to interpret.
 - We added some text to the section “Metabolic cost constraints” making this clearer. We also agree that it would be interesting to directly test for energy efficiency, and we have added some text to the discussion, including this citation, in the context of relating high level concepts (such as specialization) to low level constraints and principles (such as energy consumption).
- Figure 4 is one of the figures that can benefit from more explanation. For example, it is not clear what the vertical dashed lines represent. Maybe even a color bar to have a rough idea of which p corresponds to which color. But that is not really necessary since I believe the trend is more important than the actual values.
 - We agree and have added a color bar and some explanatory labels that hopefully make it clearer.
- The authors mention the often neglected dynamics of modularity in the second line of the discussion. I agree with their statement, but I think, for completeness, they should mention

that there are works on this topic and the methods are there in case the reader wanted to follow them up. Here are two seminal examples:

- Shine, J. M., Bissett, P. G., Bell, P. T., Koyejo, O., Balsters, J. H., Gorgolewski, K. J., Moodie, C. A., & Poldrack, R. A. (2016). The Dynamics of Functional Brain Networks: Integrated Network States during Cognitive Task Performance. *Neuron*, 92(2), 544–554. <https://doi.org/10.1016/j.neuron.2016.09.018>
- Bassett, D. S., Wymbs, N. F., Porter, M. A., Mucha, P. J., Carlson, J. M., & Grafton, S. T. (2011). Dynamic reconfiguration of human brain networks during learning. *Proceedings of the National Academy of Sciences of the United States of America*, 108(18), 7641–7646. <https://doi.org/10.1073/pnas.1018985108>
 - We agree and cite it accordingly in the discussion.
- I wonder if the authors explored the excitatory/inhibitory ratio of the inter-module connections? Based on the previous works on circuit mechanisms of perceptual decision-making, I expect them to be mostly inhibitory, at least in the case with separate outputs. Also, I think with increasing sparsity, the network should prioritize inhibition between modules so the ratio should get to larger values. The authors are welcome to conduct the analysis, but I don't see it being essential for their story, so they can also speculate this in the discussion if that's too much to do.
 - We were very interested in this suggestion, but unfortunately, we hadn't stored all the precise weights from every run featured in Fig. 3, and we didn't want to rerun everything, so we reran a smaller sample of parameters and didn't see any obvious pattern (see figure below).

Inhibitory to Excitatory ratio of both internal connections and between modules communications, depending on the sparsity of communications (p).

- Another possible extension is to add more modules than two. E.g., what happens if there are three modules? Will the third module divide into two or the allocation would be asymmetric given some feature in the input? Would 5 modules just make the network more robust by providing a backbone for redundant computation? Again, not essential but I'm curious to at least have the authors' thoughts on this. I feel these two points (10 and 11) can potentially expand into a separate paper, exploring the dynamics of interaction and resource allocation in modular neural networks.
 - We agree that it would be interesting to extend to multiple modules, and that this is beyond the scope of this paper. In terms of our thoughts on this: expanding to a third module would require a task that had three relevant subtasks in the same way that the current task has two relevant subtasks. Obviously at some level this is the interesting question that the brain has to tackle, however in terms of scientific understanding, doing this in a clean way that allows you to draw clear conclusions is very challenging! We have started to look at more challenging real-world versions of these tasks such as decision-making dynamics in an asymmetrical setup (two modules doing a sort of pre-processing of the input, while the third would be tasked with decision based on the parity), including with a new PhD student in our group, but as yet it is too early in the project to draw any strong conclusions.

- I wish the authors the best of luck in their future works and again, well done on this interesting project.
 - Many thanks! 😊

Reviewer #3 (Remarks to the Author):

This paper studies the factors leading to the emergence of functional modularity in artificial neural networks. Its main finding is that structural modularity is not a sufficient factor (except at extreme values). Hence, the Authors conclude that, similarly, functional modularity in animal brains may not hinge entirely upon the brain structure.

This paper poses an exciting question and provides a solid motivation for its experiments. Its findings are non-trivial and help shed light on the very notion of modularity. Nevertheless, this paper also suffers from a series of limitations. Its conclusions are too broad as they are based on a single task (parity-based digit classification) and a single neural architecture (recurrent neural networks). At the same time, the formal description of such neural architecture is not fully satisfactory as it would not be possible to reproduce it based on the paper alone. Finally, the paper does not reference a large portion of the literature on this topic from the field of machine learning, which provides evidence that multi-task learning and dynamic, varying environments are the key to functional specialisation.

- We address each of these comments in reply to the more detailed comments below.

In light of both the strengths and weaknesses highlighted above, I recommend that this paper is accepted only after major revisions to address my comments below (divided by section).

Abstract

- The abstract contains a conclusion (1) that does not seem to be warranted given the experiments reported in the paper, namely that "specialization can only emerge in environments where features of that environment are meaningfully separable". All the inputs consist of separable variables (two digits) so no control experiments are present.

The control experiment is the case where we introduce correlation (c) between the digits. In the extreme case $c=1$ the two digits will always be identical, while when $c=0$ the two digits are independent. In less extreme cases, knowing one digit will give some information about the other digit. The results of varying c can be seen in Figure 3. Higher values of c lead to little or no specialization. However, we have modified the abstract to make it clear that we only make this statement in our particular setup.

- The phrasing of the conclusion (3) in the abstract also seems too general: "findings are qualitatively similar across different network architectures" makes it sound like different neural architectures were tested (such as MLPs, CNNs, Transformers, etc.) whereas only (variants) of recurrent neural networks are included in the paper.
 - We agree and have modified the abstract accordingly to reflect this.

Introduction

- Here the Authors make a compelling case for using in-vitro analyses of modularity based on artificial neural networks, given that biological neural networks can only be studied imperfectly through lesions or brain imaging. As a person expert in machine learning but not in neurosciences, I found this section easily digestible by a wide array of potential readers.
- The Achilles' heel of this section is its lack of references, especially from the field of AI and machine learning, of previous work that attempted similar analyses on artificial neural networks.

Some of these works employed tools similar to this paper's experiments. For instance, Meyes (2020) conducted ablation analyses on subpopulations of neurons.

- We have added a number of references to the introduction in response to the comments of the reviewers, including this one.

Other missing references highlight some of the limitations of the present work, which I discuss below.

- ***On measuring structural modularity*** Structural modularity was found to emerge even in fully connected neural networks (Watanabe 2019, Casper et al. 2022). In practice, when a connection is present but its weight is approximately 0, it becomes equivalent to a missing edge in the network. Hence, measuring the level of structural modularity based on the Q-metric, which ultimately depends on the ratio of cross-module connections p , may be insufficient.
 - Thank you for pointing us to these two papers, which we have added to the introduction. We tried to use the measure from Casper et al. (2022) but were unable to obtain meaningful results for our networks. We are not entirely convinced by these approaches and, although interesting, they do not seem as well accepted in the literature as the Q metric. As an example of our concerns about this approach, a small weight value doesn't necessarily mean a small effect. For example, if the other incoming weights to a neuron are also small and the output weights from that neuron are large, or if there are many small weights from neurons with similar selectivity. In any case a biological synapse with a small weight is still physically present and would appear on connectivity-based measures of modularity used in neuroscience, as well as contribute to the metabolic costs of the network.
- ***On the variables leading to the emergence of functional modularity*** The emergence of functional specialisation in modules is known to be driven by multi-task learning (Yang et al. 2019, Dobs et al. 2022) and dropout regularisation (Lange et al. 2022), among others. This illustrates a fundamental limitation of this work, which focuses on a single (global) task. In fact, modular behaviour may emerge under different structural conditions when multiple global tasks, composed of different sub-tasks, are taken into consideration.
 - We did not mean to suggest that modularity could only arise in the conditions we analysed. We agree that there are a number of things one can do to encourage the emergence of modularity, including changing the learning rules, environment and task structure, training regime and network architecture. Rather, our aim was initially to investigate whether or not structural modularity was sufficient by using a carefully controlled setup, and then to investigate which features of this carefully controlled setup were required to promote modularity. We have added some additional text to the Discussion to make this point clearer, as it was raised by multiple reviewers.
- ***On defining compositional generalisation as a criterion for modularity*** Fodor's first principle hints at a definition of modularity as the ability to *re*-compose modules in new combinations, which leads to systematic generalisation in new global tasks composed of known sub-tasks. For instance, Csordas et al. (2021) showed that this is not the case in vanilla artificial networks: modules corresponding to different sub-tasks are not re-used for similar sub-tasks and are not composed for global tasks requiring new combinations of sub-tasks. The present paper should consider discussing functional modularity from a compositional perspective, considering systematic generalisation as one of the analysis tools. Mittal et al. (2022), for instance, showed how structural modularity through routing achieves better generalisation.

- We agree that generalisation and re-composition of modules is an important aspect of modularity that we didn't touch in this study. Unfortunately, this would not be possible to investigate quantitatively in the context of our controlled model setup. However, we agree that it is important to discuss this point and we have added some text to the introduction and discussion, as well as citations to these extremely relevant papers.

Methods

- The formal description of the neural architecture is lacking as it does not provide several details necessary to reproduce it. For instance, the authors mention that it consists of a "recurrent neural network". However, it does not specify: 1) what kind (a vanilla RNN, a GRU, an LSTM)? 2) how deep (i.e., how many layers)?
 - We agree and have added additional details in the Methods section to clarify.
- Moreover, it is not clear from the description how connections between modules operate. Are these connections from some dimensions of the input of layer I of module A to the output of layer I of module B? This should be clarified as this is a crucial component of the experimental setup.
 - The networks are all shallow (single-layered) RNNs. We have made this clearer, and explained this choice as it relates to the next point the reviewer is making.
- The very choice of RNNs as neural architectures is dubious: since the data (pairs of digits) are not sequential, why not use an MLP or a CNN instead? Except for the experiments in §3.3, where information is gradually unveiled, the input is identical for all time steps in the current setup.
 - We agree that an RNN is not the obvious choice of architecture for a static image task but there were a few reasons for this choice. Firstly, biological neural networks are usually recurrent. Secondly, although it is possible to define the Q metric for a feedforward architecture, it has some less-than-ideal properties that are shown in the diagram below. The first layer after the inputs must have the correlation measure of functional specialization equal to 1 because each group of neurons receives inputs only from one of the two digits. The second group can have functional specialization varying from 0 to 1, but this group on its own has $Q=0$ because there are no interconnections between the groups. Considering both layers together, you would have correlation-based functional specialization always at least $\frac{1}{2}$ and the Q-metric between $\frac{1}{4}$ and $\frac{1}{2}$. The fact that you can't make $Q=0$ can be seen because in the RNN we can have all-to-all connectivity for zero modularity, but this can't be true with a layered feedforward architecture. The third reason is that using an RNN allows us to use a consistent architecture throughout rather than switching to a different architecture for 3.3. We added a comment on this choice in the paper at the end of 2.2.

- The decision process for separate readouts is also unclear. What is the output of the readout modules? There are glimpses of information ($d_{out} = \text{argmax } r_{out}$) that make it sound like it is a probability distribution over a categorical variable (parity-based choice). But then what does it mean that r_{out} is chosen via a max function? Are you picking the module with the highest probability mass placed on any digit? Again, the lack of clarity hinders a potential reader from fully appreciating your experimental setup.

 - We agree that this section of Methods needed more details and clarity, and we have changed it accordingly.
- The subsection on metrics was well written and all the three considered metrics are well motivated in light of Fodor's criteria for modularity. What I found confusing was the name of the first metric, "Bottleneck retraining". If you are freezing and then training a new, separate network (as common in probing techniques) then you are not *re*training. Also, it is not properly a bottleneck as it directly predicts the output rather than constraining information in intermediate layers. Consider renaming this metric to "probing of the trained modules" or similar.

 - We agree and have now called this metric "Module Probing", which is indeed clearer. The text and figure have been changed accordingly.
- Another possible criticism of the third metric ("hidden state correlation") is that non-linear neural networks may encode the same function through different parameterisations (which also results in different hidden states). In other words, functional similarity does not necessarily imply parametric similarity. Consider at least measuring linear CKA in lieu of Pearson's correlation as it captures similarity invariant to orthogonal transformations.

 - We agree that in a general case of comparing neural networks, using the CKA is generally a good practice to measure similarity. However, in our case we are only comparing module activity to itself, so we don't need to correct for orthogonal transformations, and we therefore opt for the simpler Pearson correlation. We added a discussion of this point to the Methods.

Results

Overall, this section has an in-depth discussion of the different factors that may affect modularity, which is stimulating.

One limitation is the discussion of architecture choices and in particular output schemes. If anything, the bottleneck variant where modules compete for readout creates an inductive bias *against* modularity as defined in this task, because modules would tend to specialise towards subsets of *examples* rather than different subsets of variables within each example.

- Unfortunately, we are not entirely sure we understood this comment. Our results show that the bottleneck variant leads to more modularity, demonstrating how hard it is to develop reliable intuitions when it comes to such questions.

References

Stephen Casper, Shlomi Hod, Daniel Filan, Cody Wild, Andrew Critch, and Stuart Russell. Graphical clusterability and local specialization in deep neural networks. In ICLR 2022 Workshop on PAIR2Struct, 2022.

Katharina Dobs, Julio Martinez, Alexander JE Kell, and Nancy Kanwisher. Brain-like functional specialization emerges spontaneously in deep neural networks. *Science Advances*, 8(11):eabl8913, 2022.

Lange, R. D., Rolnick, D. S., & Kording, K. P. (2022). Clustering units in neural networks: upstream vs downstream information. arXiv preprint arXiv:2203.11815.

Meyes, R., de Puiseau, C. W., Posada-Moreno, A., & Meisen, T. (2020). Under the hood of neural networks: Characterizing learned representations by functional neuron populations and network ablations. arXiv preprint arXiv:2004.01254.

Mittal, S., Bengio, Y., & Lajoie, G. (2022). Is a modular architecture enough? *Advances in Neural Information Processing Systems*, 35, 28747-28760.

Chihiro Watanabe. Interpreting layered neural networks via hierarchical modular representation. In *Neural Information Processing - 26th International Conference, ICONIP 2019, Sydney, NSW, Australia, December 12-15, 2019*, pp. 376–388.

Guangyu Robert Yang, Madhura R. Joglekar, H. Francis Song, William T. Newsome, and Xiao-Jing Wang. Task representations in neural networks trained to perform many cognitive tasks. *Nature Neuroscience*, 22(2):297–306, 2017.

Reviewer #4 (Remarks to the Author):

Thank you for this very interesting paper.

Please find a formatted version of the review in the attached pdf.

Review of “Dynamics of specialization in neural modules under resource constraints”

Based on experiments with a modular (2 pathway architecture in 3 different configurations) the paper makes the following claims

1. Structural modularity isn't sufficient to lead to functional specialization unless there are extreme resource constraints.
2. Correlated inputs lead to correlated states (i.e., disfavor specialization)
3. Temporal dynamics of specialization are relevant and should be studied more.

I agree that modularity is an extremely important topic for the neuromorphic field as the connectivity is constrained to a topology with strong local, but sparse global connectivity, like a small world topology. In addition, it looks like the topic of modularity is understudied, especially from a neuromorphic perspective.

Unfortunately, it is not clear how the conclusions of this paper are meaningful for neuroscience or neuromorphic engineering even though this seems to be the motivation for the paper. The discussion on the relevance for neuromorphic approaches seems to be an appendage and rather vague.

We do not agree with this assessment, and our impression is that many researchers working in the fields of neuroscience and neuromorphic engineering that we have presented this work to have found it valuable. Indeed, the other reviewers of this paper seem to agree, for example reviewer 2 states that it “has major implications for both neuroscientists aiming at understanding cognition from the brain’s intricate network structure, and for AI researchers aiming to incorporate biologically plausible architectures.” We respond to the more detailed comments inline below.

Additionally, some claims are a bit questionable. Claim 1. seems to depend strongly on the specific choice of task, architecture, and definition of modularity. Claim 2. seems almost trivial given the way specialization is measured here and claim 3 is vague.

Claim 1 states that structural modularity is not sufficient to lead to functional specialization. To establish this, a simple example where it does not do so is enough. We do not make a stronger universal claim that structural modularity is never relevant. We have added some text to the discussion to make this clearer, as well as modifying the abstract.

Claim 2 may seem obvious in our simple and controlled setup, but it is not so clear in more general settings. Indeed, in the real world correlated inputs are very common and this leads to confusion both for neuroscience and machine learning. How you approach answering these questions is very consequential for how you define and study specialisation.

Claim 3 is vague in that we do not propose a complete answer, but we believe it raises an important question that has been insufficiently studied.

Nevertheless, I appreciate the originality and simplicity of the approach as it can lead to direct and easy to understand conclusions on a very complex topic (as long as we are aware of the risks of such simple conclusions).

Additionally, one cannot expect a single paper to be comprehensive on such a complex topic, but a paper that aspires to be published in such a renowned journal and makes broad claims should try to address some of these issues (see more detailed info below).

So, if revised accordingly and significantly, this paper might serve as a starting point for a deeper study of modularity from a neuromorphic perspective.

Do the conclusions of this paper generalize to more complex architectures?

Our approach in this paper was to investigate whether or not architectural features alone could give rise to modularity, because this appears to be an implicit assumption made by many researchers when trying to design or understand networks and modularity. We have modified the text (particularly the discussion) to make clear that we don’t claim that the only way to get modularity is from extreme sparsity and resource constraints, and that there may be other architectures, learning rules, training regimes, etc. that can also promote modularity in different architectures. Our point is simply that we shouldn’t expect it to emerge simply because we have a structurally modular architecture.

The topic of modularity is very complex, as there are many forms of modularity and specialization (fused/hybrid networks, clusters within networks, parallel pathways, sequential blocks, modularity within nodes, etc., see Amer and Maul 2019). As far as I can tell, all types of modularity can also be observed in the brain on different levels, but only one definition is investigated here.

This already indicates that the simple conclusions that are drawn in this paper are only limited to a specific case and might not have the general relevance that is implicitly claimed.

We did not wish to make any such implicit claims, and have made this clearer in the revised abstract, introduction and discussion.

The most important limitation in my opinion is that the paper only looks at modularity related to the input and the output of the network, but the more interesting and important modularity arises due to the task and the structure of the hidden layers. Especially modern neural network architectures are quite modular by design. For instance, transformers have a multi-head structure, and each head contains specialized parts.

We disagree with this summary of our work. Both the task and architecture / hidden layer structure in our study are designed to be modular in a precisely controllable way. Just as a transformer has an architecture that would appear to favour specialisation, our network has a structure that would appear to favour specialisation (for example, when the majority of the inputs to one of the recurrent layers comes from only one of the two images) but surprisingly, we find that in many cases specialisation does not emerge despite this. We agree, however, that there are other forms of task modularity and other forms of architectural modularity (as well as different learning rules, training regimes, etc.), and that specialisation may emerge in some of these. We have added some text to the discussion on this point.

Particularly, the paper claims that structural modularity is not sufficient to guarantee functional specialization, unless there are extreme constraints. The multi-headed transformer architecture seems to prove this wrong. In transformers, the multi-headed structure leads to functional specialization via the attention mechanism (which I would consider part of the structure) without extreme resource constraints.

As discussed in response to comments above, we showed that structural modularity is not a sufficient condition to guarantee specialisation. If it were, our structurally modular networks would give rise to specialisation, and they do not. However, we do not claim that there are no architectures in which specialisation arises due to other features (say, learning rules, training regimes, etc.). We have made this clearer in the revised discussion.

Of course, one could argue that there are constraints in the number of neurons per head or the number of heads, which are relevant due to the huge size of the input and output spaces and the complexity of the task.

It is however also possible that, on the contrary, mechanisms like attention may be needed to make use of modularity and the conclusions in this paper simply cannot be applied to such architectures. I am not sure if this question can be answered based on the currently available evidence.

We would be delighted if readers were to take home the message that additional mechanisms beyond just structural modularity are necessary for specialisation to emerge. This is precisely what we argue, and we have made this clearer in the revised version.

A number of papers (that are ignored in this work) show that neural networks are in fact quite modular (even though they cannot make good use of this modularity). For instance:

Csordás et al. 2021 provides a compelling way to measure modularity based on the weights and <https://arxiv.org/pdf/2010.02066.pdf>

Filan et al. 2021, Clusterability in neural networks <https://arxiv.org/pdf/2103.03386.pdf>

Hod et al. 2022, Quantifying local specialization in deep neural networks <https://arxiv.org/pdf/2110.08058.pdf>

Zhang et al. 2023, Emergent Modularity in Pre-trained Transformers

So, it is unclear if the claims of this paper are limited to the chosen simple toy architecture. This should be discussed or addressed by additional experiments.

Our main aim was not about the emergence of structural modularity, which we control directly in this paper, but rather to relate structural and functional modularity. The point that these networks demonstrate structural but not functional modularity is in line with our results. We also agree that these works are relevant, and we have added some text including these citations to the Discussion.

An even more interesting aspect of modularity, that is essentially ignored in this paper (and also implicitly questions the paper's claims) is the effect of modularity on generalization. Here is a small selection of papers that address this topic:

Goyal et al. 2021, Recurrent independent mechanisms, <https://arxiv.org/pdf/1909.10893.pdf>

Kirsch et al. 2018, Modular networks: Learning to decompose neural computation <https://proceedings.neurips.cc/paper/2018/file/310ce61c90f3a46e340ee8257bc70e93-Paper.pdf>

Bahdanau et al. 2019, Systematic generalization: what is required and can it be learned? <https://arxiv.org/pdf/1811.12889>

While most of these papers construct modular models as an inductive bias, it seems plausible that if a task requires generalization capabilities, architectures could become modular through some form of optimization (be it evolutionary or gradient-based). I.e., there are (structural and non-structural) drivers of (structural and functional) modularization beyond resource constraints. (An interesting aspect to investigate, of course, could be, if resource constraints and generalization could go hand in hand and constraints can lead to better generalization)

We absolutely agree that modularity is most interesting in the context of re-use and generalisation. Indeed, we got interested in the study of modularity in the first place precisely for this reason. We have solely focused on functional specialization here as the first step towards understanding modular composition of knowledge at every level: structure, function and composition. What this paper shows is that going from structure to function isn't as straightforward as one might think. Csordás et al. (2021) shows that going from functional specialization towards meaningful composition isn't straightforward either. We also completely agree that other tasks, environments and learning rules can promote modularity (and probably do). We have added some text on generalisation and re-use to the introduction and discussion, as well as citations to these specific papers.

And here is yet another example where modularity seems to be useful due to task requirements:

Ellefsen et al. 2015, Modularity to avoid catastrophic forgetting, <https://doi.org/10.1371/journal.pcbi.1004128>

We agree, and we had already cited this in the discussion, in the context of the roles of varying environments and multi-task learning.

Are the conclusions of this paper relevant for neuroscience?

The previous criticism concerning the non-applicability to more complex architectures certainly applies to the brain. It is, in fact, fairly easy to come up with a counterexample to the claim that "structural modularity isn't sufficient for functional specialization" (ok, I admit there is a chicken and egg problem here):

We do not understand this. You can't have a counterexample to a claim that a universal statement is not true. To counter our claim that structural modularity does NOT imply functional specialisation you would have to prove the universal statement that structural modularity DOES imply functional specialisation.

Efficient modulation in the brain by “spraying” a small region (structural module) with transmitters (e.g., dopamine) would fail if there was not also a certain extent of functional specialization.

This, and the comments below, seem to address a claim that we do not make. We are not saying that structure and function are entirely decorrelated in the brain. Indeed, we start by noting in the introduction that the reason people think that structure and function are related is because they seem to be in the brain, as found right from very early studies.

I.e., if we assumed the structure as fixed, functional specialization would likely have to arise through optimization (so, for complex enough structures, structural modularity is (likely) sufficient for functional specialization).

This actually agrees with our point. If you assume that structure is fixed, then it must be something other than structure that is promoting specialisation. We have shown that in the case of our task in which specialisation is possible, standard training with structural modularity is not sufficient for functional specialisation. For different sorts of training, it may be.

Another example, that is closer to the idea of the paper might be the following: Let’s say we have a convolutional-like architecture that resembles visual cortex and an architecture that is designed to detect different temporal frequencies. Both start with random parameters and receive both visual and auditory input (let’s say spoken and written MNIST). Would we really assume, based on this paper, that the modules would not specialize on the respective suitable input given by the structure?

Structural modularity, as we have defined it, and consistent with a lot of literature, especially neuroscience literature around connectomics, has a precisely defined meaning in terms of graph connectivity as measured by the Q metric. The claim “structural modularity is sufficient for functional specialisation” means that if the Q metric is high enough, functional specialisation must emerge. This is what we have shown to be incorrect. We entirely agree that other measures of structure may have different properties in terms of specialisation.

With that said, this would be a really interesting idea to test. We do not think it is appropriate or necessary to speculate on the results of such an experiment but at least to us it is not as obvious as the reviewer suggests. For example, state of the art speech recognition models do often feature a convolutional frontend, e.g. Wav2Vec2.0 (Baevski et al. 2020).

Additional comments

1. I assume the classification accuracy for the task is not reported as it does not play a role for modularity, but it seems relevant to at least mention it. Is it always 100%?

The presented networks are likely strongly overfitted to the task (there is no mention of a training, validation, or test set). This might have an influence on the given measures, as the level of training could affect modularity.

Especially in the extremely resource constrained cases, it matters if the networks are trained to 100% accuracy or with a fixed number of epochs (which might lead to lower accuracy). In the former case, I expect more specialization than in the latter. So, to be able to really understand and put the results in context, reporting the accuracy is needed.

We always test on a separate testing set, and train for a fixed number of epochs to avoid overfitting. Across the full range of values of p , test accuracy varies from around 60% to almost 90%, so never at chance or saturation. The task is therefore not trivial for networks of that size, which was one of our objectives as to not over-parameterize the networks and make the results meaningful.

We added this plot to the supplementary materials and discussed these points in the main text.

*Accuracy of networks with 25-neurons modules for varying sparsity (p) values.*

2. “In this structural configuration, networks often resort to having a main decision-making module. Although the task is designed to be solvable in a modular fashion, having such a module take all decisions means it’s able to predict both digits and thus displays less specialization.”

This seems a bit contradictory. If there is a main decision-making module, it seems like there is a form of specialization, it is just not detected by the measure.

We agree that our metric ignores this potential definition. We have modified the text in the results to reflect this.

3. The paper makes some assumptions about the structure of how the networks solves the task that do not seem plausible or are at least not proven.

Section 2.2.1 talks about “sub-tasks” and a “global task”. This makes sense from a human perspective as that is how we would approach such tasks, but it is not shown that the network actually solves the task in such a manner. On the contrary, I think there is no reason to believe that. It will likely just find a shortcut. Instead of classifying digits separately, it can just learn directly which pixel combinations correspond to which output (in a nonlinear fashion, of course).

We agree. When specialisation drops, it probably is using a shortcut, and that’s why it’s interesting to look at scenarios where there is low bandwidth between the modules and so raw pixel data is hard to share. On the other hand, when a module is probed and there is no way to pick up on the other digit, we can confidently say that not enough pixel data is passed on at the communication level. (In other words, as soon as modules can shortcut, they probably do).

4. “We hypothesize that this is a result of the static nature of inputs, where all the information is readily available at every time step and so there is no advantage to ongoing communication after the initial burst.”

I do not fully agree with this interpretation. There is still “communication” between the networks as they are still connected. The other module will likely not create a memory of the opposite module as it can just read it out at each timestep. I.e., the specialization measure does not allow to judge communication. To show that one module develops a memory of what happens in the other, one would have to cut the connection in timestep 3 or 4 (and maybe run it longer).

I think that the fact that the modules have to create a memory of the opposite side (as they cannot rely on the input) may be the reason why one can observe overall lower specialization in the noisy input case.

This is an excellent point, and we have modified the text to discuss this. In the noisy version however, modules depend on averaging out the noise to consistently predict the output, so there is extra advantage to ongoing communication in the noise case. That, however, does not negate the existing advantage of maintaining communications to free-up working memory even in the noise-free version.

5. Please also note the following recent paper that seems to be relevant:

<https://www.modulardeeplearning.com/>

<https://arxiv.org/pdf/2302.11529.pdf>

We already cited this paper in the introduction (end of second paragraph of subsection “our approach”).

Minor comments to improve form, language, understandability, etc.:

1. The language is good, but for punctuation and to reduce informal and wordy language it might still help to use a grammar/language tool. An example for wordiness is: “such as audio-visual integration for example”

We have substantially rewritten several parts of the text to improve clarity.

2. Typo: “and is run”

Fixed

3. Equation numbering takes a break after (4)

Fixed

4. Fig. 3: color bar should be labelled; The colormaps, at least in each lettered subfigure should have the same range. I would prefer all Figures to have the same range even if that makes it harder to see the small differences in subfigure C.

We agree, and fixed it using a log color scale.

5. Fig. 4: Please label axes and add a colormap. The caption mentions the meaning, but it makes it much harder to read the figure. The colors are only explained in C. but mentioned in A. I would also not make the line dashed; this is confusing and unnecessary as the lines are clearly separable.

Fixed

6. Typo: “and is run for a number of time steps”

We reworded this.

7. Please define % (modulo) and D1, D2 (numerical value of digits?) in Eq.1.

Fixed

8. I would suggest to first explain the task, then the networks, as it is easier to understand the network structure once it is clear what the input and output are.

We agree, and we have now swapped the order of these subsections.

9. The part about “Metabolic constraints” (bottom of page 7) is not explained well. Also, you should refer to Fig.3 in this paragraph.

We have improved the text and added the reference.

10. Please explain better what the outputs of the network are. Especially explain what the difference between separate and fused is.

We have added clarifications regarding the output.

11. "Functional specialization is usually understood as a static property" Please provide a reference for this statement.

This is not explicitly stated, but is an implicit assumption in almost all the papers we have looked at that study functional specialisation. Citing a single paper for this statement would feel out of place, although in the discussion we do cite the exceptions that explicitly include temporal dynamics into their definitions of specialisation (e.g. Fakhar et al. 2023). We have reworded this sentence to make it clear that it's an implicit assumption.